# Deep active optimization for complex systems

Ye Wei [1,2,3,10] ✉, Bo Peng [4,10], Ruiwen Xie[5,10], Yangtao Chen[6,10], Yu Qin[7,10], Peng Wen[4,10], Stefan Bauer[6,10], Po-Yen Tung[8,10] & Dierk Raabe [9] ✉

Inferring optimal solutions from limited data is considered the ultimate goal in scientific discovery. Artificial intelligence offers a promising avenue to greatly accelerate this process. Existing methods often depend on large datasets, strong assumptions about objective functions, and classic machine learning techniques, restricting their effectiveness to low-dimensional or data-rich problems. Here we introduce an optimization pipeline that can effectively tackle complex, high-dimensional problems with limited data. This approach utilizes a deep neural surrogate to iteratively find optimal solutions and introduces additional mechanisms to avoid local optima, thereby minimizing the required samples. Our method finds superior solutions in problems with up to 2,000 dimensions, whereas existing approaches are confined to 100 dimensions and need considerably more data. It excels across varied real-world systems, outperforming current algorithms and enabling efficient knowledge discovery. Although focused on scientific problems, its benefits extend to numerous quantitative fields, paving the way for advanced self-driving laboratories.

Modern society benefits tremendously from superior solutions in engineering control systems, materials science, physics, biology and computer science. These advancements improve infrastructure, healthcare and technology, enhancing quality of life and addressing global challenges. Examples include use of advanced engineering control for autonomous systems, the discovery of high-performance alloys for better and more sustainable building materials and the development of life-saving pharmaceuticals, including drugs optimized to combat diseases such as COVID-19[1]. However, identifying such superior solutions is challenging due to the vast size and often highly nonlinear nature of the search space.

Moreover, conducting experiments or simulations can be extremely costly, with processes such as synthesizing and characterizing advanced alloys or drug-relevant molecules often costing millions of dollars and taking months or even years of intense labor.

Optimization performed by human experts typically relies on educated trial-and-error navigation of the search space, often leading to substantial expenditures of both resources and time, particularly in cases of highly nonlinear interactions. The rise of artificial intelligence (AI) offers a powerful alternative that can minimize human bias and achieve better solutions at minimal cost. Unlike traditional optimization algorithms, which are generally assessed on the basis of function evaluations, these data-driven AI algorithms operate in a closed loop to guide experiments or simulations, iteratively identifying and labeling the most informative data points to discover the next best candidates while minimizing data labeling efforts. This approach is known as active learning (AL)[2–4], and there has been a surge of interest in developing AL-based self-driving laboratory in all areas of physical, chemical and biological science[5–9].

As illustrated in Fig. 1, We designate our algorithm as active optimization (AO), which aligns closely with Bayesian optimization (BO) in

[1]Department of Data Science, City University of Hong Kong, Hong Kong, China. [2]Department of Materials Science, City University of Hong Kong, Hong Kong, China. [3]Hong Kong Institute of AI for Science, City University of Hong Kong, Hong Kong, China. [4]Department of Mechanical Engineering, Tsinghua University, Beijing, China. [5]Institute of Materials Science, Technical University of Darmstadt, Darmstadt, Germany. [6]Helmholtz AI, Munich, Germany. [7]Department of Orthopedics, Peking University Third Hospital, Beijing, China. [8]MatNex, London, UK. [9]Max Planck Institute for Sustainable Materials, Düsseldorf, Germany. [10]These authors contributed equally: Ye Wei, Bo Peng, Ruiwen Xie, Yangtao Chen, Yu Qin, Peng Wen, Stefan Bauer, Po-Yen Tung. ✉e-mail: ye.wei@cityu.edu.hk; d.raabe@mpie.de

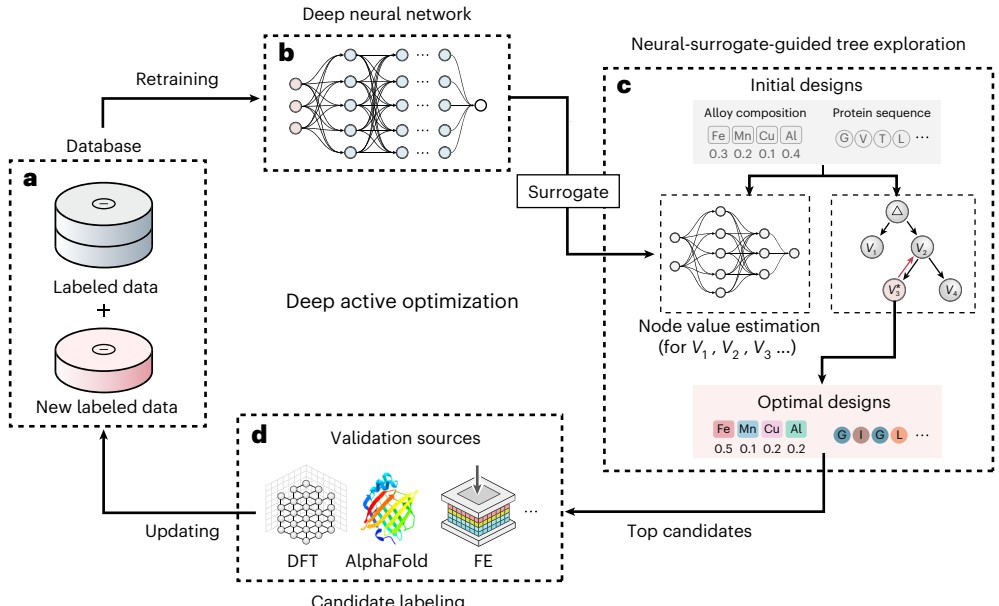

**Fig. 1 | Deep AO with neural-surrogate-guided tree exploration (NTE).**
**a**, Database of the complex system of interest. **b**, DNN that learns the input–output relationship. **c**, NTE uses the DNN as the surrogate model to find the optimal designs. Here, alloy compositions and protein sequences are used as examples, starting from random initial designs and converging to optimal ones. **d**, The validation source provides ground truth for the top candidates. Here, three examples are used: FE methods, DFT and AlphaFold.

terms of its objectives and overall framework. However, BO primarily utilizes kernel method and uncertainty-based acquisition function to identify 'optimal' candidates, whereas AO generalizes the application of surrogate models and search methodologies, allowing adaptation across a wider variety of method types, thereby enhancing its versatility and scope beyond traditional BO approaches. Furthermore, AO is akin to the AL framework but differs in terms of its goal—instead of improving the model predictivity, AO aims at finding the optimal solutions with a relatively small initial dataset (from a few dozen to hundreds).

It is well accepted that the knowledge of the internal interactions inherent in many complex systems (validation source) are usually not fully accessible and the structure, gradient and convexity of the objective function are unknown[10]. Therefore, a surrogate model is often used to treat such nonconvex, nondifferentiable systems as a 'black box' and approximate the solution space of the complex system through a learning model[11,12]. Some machine learning (ML) models, such as Bayesian methods, heavily rely on assumptions about prior distributions or feature engineering[13–15], while others, such as decision trees, are prone to overfitting and are limited to processing specific data types, such as tabular formats. Consequently, they often struggle to accurately capture intricate relationships and dependencies in high-dimensional big datasets, leading to poor generalization in unseen scenarios and slower convergence in high-dimensional spaces[16,17]. The advancements of deep neural networks (DNNs) present a compelling alternative for approximating high-dimensional nonlinear distributions of any data type[18,19], and the effectiveness of this approach is indicated by its remarkable accomplishments across various fields, including image classification, natural language processing and autonomous vehicles[20,21].

Another approach that could identify optimal solutions within complex systems is the so-called reinforcement learning (RL), which is defined as an ML algorithm that searches for optimal solutions through interactions with an environment. However, RL differs from AL in three major aspects, as it often requires (1) easy access to reward functions, (2) numerous training data and (3) cumulative reward. In particular, the RL that combines DNNs with the Monte Carlo tree search (MCTS) method has demonstrated remarkable success in such tasks, particularly when large datasets are accessible and cumulative objectives are considered.

Its tremendous success is exemplified by the superhuman performance of AI players such as AlphaGo, AlphaZero and AlphaStar in various board and strategy games[22–24]. Despite these considerable advancements, combining DNN with tree search methods to tackle complex problems with limited data availability and noncumulative objectives remains elusive. This challenge arises from two primary factors:

(1) RL generally needs extensive access to reward functions or large datasets for training, whereas real-world problems often have limited, costly-to-collect data, making it difficult to train effective policy networks[25,26].
(2) MCTS is mainly suited for cumulative reward maximization in sequential decision-making and is less naturally adapted for noncumulative objectives, despite its success in superhuman AI[27–29].

Nevertheless, recent studies have utilized MCTS to iteratively partition the search space and select solutions based on upper confidence bound (UCB) and classic learning models. These methods encounter challenges when addressing high-dimensional, nonlinear distributions[30,31]. The number of partitions in high dimensions increases exponentially, and the local models struggle to generalize to the complex distribution, resulting in suboptimal performance in these tasks[32].

In this work, we introduce deep active optimization with neural-surrogate-guided tree exploration (DANTE) for the accelerated discovery of superior solutions to real-world systems characterized by limited data availability (initial data points ~200 and sampling batch size ≤20) and noncumulative objectives. Our pipeline is rather general, capable of addressing a wide range of scenarios. The pipeline begins with a database used to train a DNN, which serves as a surrogate model. Subsequently, a proposed tree search, modulated by a data-driven UCB (DUCB) and the DNN, is used to explore the search space of the complex system through backpropagation method (Fig. 1). Top candidates are sampled and evaluated using validation sources, with the new labeled data being fed back into the database (Fig. 1d).

We benchmarked DANTE against various AL algorithms to evaluate its performance across these diverse settings. First, we evaluate DANTE's

performance across six easily computable nonlinear synthetic functions with known global optima, covering dimensionalities ranging from 20 to 2,000. DANTE consistently outperforms all state-of-the-art (SOTA) methods in these tests, achieving the global optimum in 80–100% of cases while using as few as 500 data points. Second, we assess DANTE on real-world problems across various disciplines, including computer science, physics, optimal control and materials science. In these scenarios, ground-truth labels are noise-free and obtainable at a reasonable cost. Nonetheless, the search spaces are often constrained by external nonlinear conditions, adding complexity to the tasks. DANTE consistently identifies superior solutions, outperforming other SOTA methods by 10–20% in benchmark metrics, all while utilizing the same number of data points. Finally, we apply DANTE to resource-intensive, high-dimensional, noisy and complex tasks with unknown optima, such as complex alloy design, architected materials design and peptide binder design. In these cases, DANTE successfully identifies superior candidates, achieving improvements of 9–33% while requiring fewer data points relative to SOTA methods. Through extensive investigations into the learning process, we validate that the integration of deep learning and tree search is effective for discovering optimal solutions across diverse disciplines, utilizing minimal data points.

## Results

### Neural-surrogate-guided tree exploration

The neural-surrogate-guided tree exploration (NTE) is the key component of DANTE, aiming at optimizing exploration–exploitation trade-offs through a number of visits and an ML model to deal with noncumulative reward optimization problems. It resembles the setting of RL, but without the need to train an actor policy network.

NTE is inherently a frequentist's approach and uses the number of visits to facilitate the exploration–exploitation trade-off. Unlike traditional Bayesian black-box optimization algorithms, which primarily use uncertainty as the basis for this trade-off, NET treats the number of visits to a particular state as a measure of uncertainty. The more frequently a state is visited, the lower its associated uncertainty. This approach is common in MCTS-based methods. We have made some key modifications that deviate from traditional settings, enhancing our methodology's effectiveness. In the following sections, we explain the working principles of NTE and the rationale behind the introduced mechanisms.

### Conditional selection

Stochastic rollout is composed of two subcomponents: (1) stochastic expansion of the root nodes and 2) local backpropagation. The NTE algorithm performs the search by iteratively executing conditional selection and stochastic rollout until the stopping criteria are met. In the first step, the root node initiates the generation of leaf nodes, which involves applying stochastic variations to the feature vector—a process termed stochastic expansion (see 'Technical details of NTE' section in the Methods).

Figure 2d conceptually illustrates how conditional selection helps to explore the search space by addressing the 'value deterioration problem'. A search tree without conditional selection often results in lower-value leaf nodes being selected during expansion, leading to a rapid decline in value and ultimately hindering the discovery of superior nodes. In NTE, if the DUCB of the root node exceeds that of all leaf nodes, the search continues with the same root node in the next round. If any leaf node has a higher DUCB, it becomes the new root, proceeding to stochastic rollout. This mechanism encourages the selection of higher-value nodes. As demonstrated in Fig. 3b, NTE without conditional selection requires up to 50% more data points to reach the global optimum (Supplementary Fig. 1).

### Local backpropagation

In noncumulative objective problems, the aim is to find the optimal single state rather than an optimal sequence of states. Conventional backpropagation techniques update values and visitation counts along the entire search path, which is suited for sequential optimization. Meanwhile, local backpropagation updates only the visitation data between the root and the selected leaf node, preventing irrelevant nodes from influencing the present decision. This mechanism enables DANTE to escape local optima by preventing repeated visits to the same node.

Figure 2e conceptually illustrates how DANTE progressively escapes local maxima by climbing a ladder formed through local backpropagation. When DANTE is trapped in a local optimum, repeated visits to the same node trigger updates in the DUCB values of the root and neighboring nodes, generating a local DUCB gradient that helps guide the algorithm away from the local optimum. Figure 3b shows that, without local backpropagation, DANTE struggles to converge even after 10,000 data points.

### DUCB

The DUCB formula is a core component of the DANTE framework, designed to dynamically balance the exploration-exploitation trade-off. It can be expressed as follows:

$$DUCB = v_{ML} + c_0 \times c(\rho) \times \sqrt{\frac{2\log N}{n+1}}, \quad (1)$$

where $v_{ML}$ represents the value of the current node predicted by DNN. Let $\rho$ represent the ground-truth distribution, and let $c(\rho)$ be a scaling factor that adjusts based on this distribution. $N$ is the number of visits of current root node, and $n$ is the number of visits of the current leaf node. Without loss of generality, we assume that the goal is to search for the global maximum; we define $c(\rho) = \max(\rho)$. $c_0$ is a hyperparameter constant that ranges from 0.01 to 1. In the following, we provide the rationale behind those terms.

The shift from UCB to DUCB is motivated by the challenges of high-dimensional noncumulative objective problems. In these high-dimensional search spaces, the vast majority of states remain unexplored, leading to visit counts of $n = 0$, resulting in infinite UCB values. Consequently, a tree search using UCB must visit all leaf nodes at least once to obtain finite values for comparison, which imposes a high computational burden. In addition, UCB typically relies on millions of simulations to produce reliable estimates, further exacerbating the computational cost. To address this issue, DUCB modifies the original UCB formula by incorporating DNN predictions for node value estimation and adding 1 to the denominator, effectively treating all nodes as if they have been visited at least once. This adjustment ensures that DUCB consistently yields finite values for every node, eliminating the need for exhaustive stochastic rollouts at each leaf.

### Adaptive exploration

Figure 2c illustrates the adaptive exploration mechanism used by DANTE. This mechanism encourages a more aggressive exploration strategy when high-value data points are discovered in the previous iteration. Specifically, $c(\rho)$ becomes larger as high-value data points are identified, enhancing the exploration term in the DUCB formula. This dynamic adjustment intensifies exploration in promising regions, enhancing exploration of valuable areas while maintaining sufficient exploitation, thereby increasing the likelihood of identifying more superior solutions. The ablation study shown in Fig. 3b indicates that DANTE, when lacking adaptive exploration, requires 50% more data points to reach the global optimum. Figure 3c qualitatively demonstrates the efficiency of DANTE by visualizing the search history of DANTE and its ablated variants using a two-dimensional uniform manifold approximation and projection (UMAP) representation (a dimensionality reduction technique[33]) applied to Rosenbrock-100d synthetic tasks. The results clearly show that DANTE swiftly identifies the 'hotspot' region associated with the global optimum and subsequently

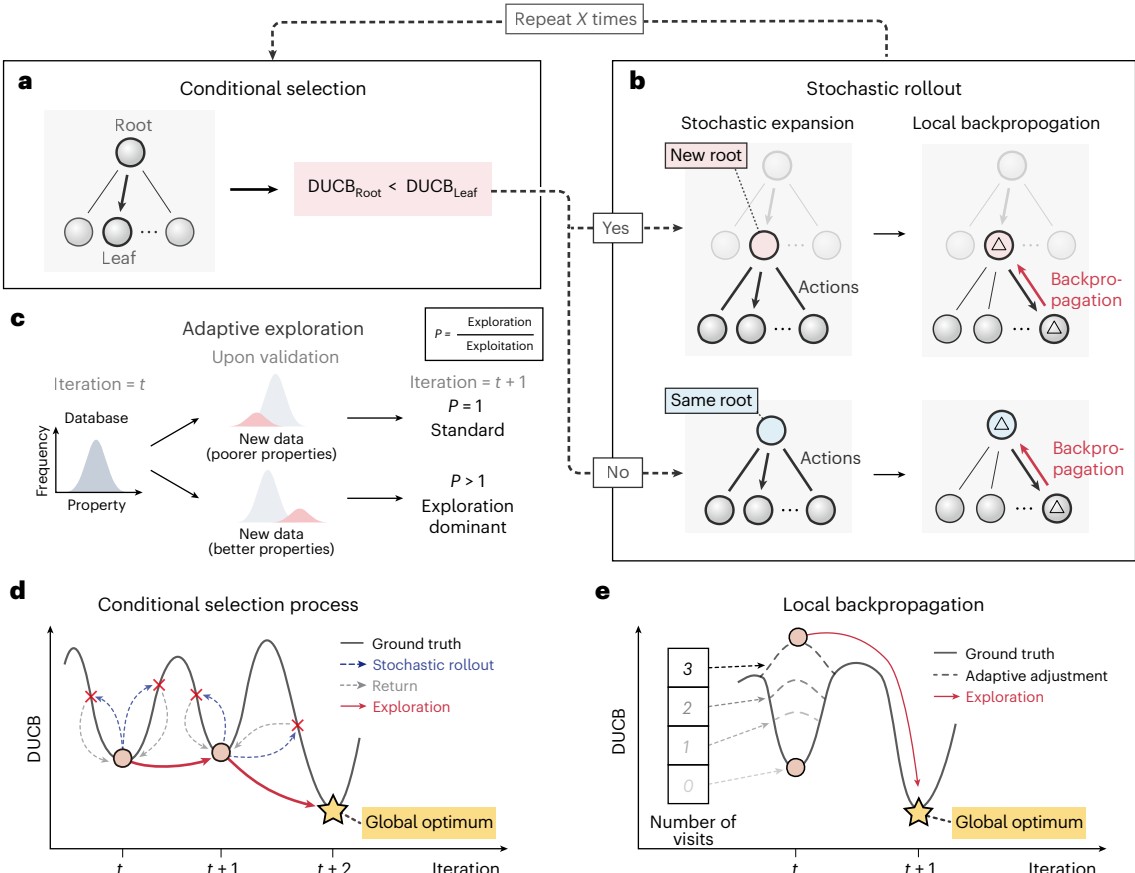

**Fig. 2 | Neural-surrogate-guided tree exploration. a,b,** Two major components of NTE: conditional selection (**a**) and stochastic rollout (**b**). **c,** The adaptive exploration introduces an exploration-dominated mode, which enhances exploration in iteration $i + 1$ when superior candidates are identified in the previous iteration $i$. **d,** The process of conditional selection. During the rollout, it compares the DUCBs of all leaf nodes with that of the root, rejecting the leaf node with the higher DUCB (represented by a red cross) and staying with the root node until a leaf node with a lower DUCB is identified. The dashed gray line indicates the stochastic rollout, and the dashed blue line indicates that the leaf node is rejected and returns back to the root in a rollout round. The circle represents the root node on the search landscape (in both **d** and **e**). **e,** Local backpropagation generates a local DUCB gradient 'ladder', helping the algorithm gradually escape suboptimal regions.

concentrates its search efforts in that vicinity. By contrast, the other variants fail to locate this hotspot, further reinforcing the findings from the ablation study.

## DNN is the key

Before undertaking expensive real-world tasks, it is crucial to benchmark various AO algorithms on both synthetic tasks with known global optima at various dimensions and low-cost, low-dimensional real-world benchmark tasks, which can offer valuable insights into the algorithm's efficiency and effectiveness across different contexts. As shown in Fig. 3a, the benchmark study demonstrates that, in comparison with traditional AO pipelines, DANTE is capable of addressing a wide range of scenarios: low- to high-dimensional problems, from easy to hard data acquisition tasks, and from simple to complex systems.

We use well-established high-dimensional, nonconvex synthetic functions with known global optima for our benchmark tests, which have been widely used to evaluate the performance of optimization algorithms. Unlike traditional optimization algorithms, where the process is often parallelizable and primarily focuses on the number of function evaluations required to reach the global optimum, our benchmark study uses these synthetic functions to mimic the complex data distribution generated by various validation sources. Our aim is to assess the number of data points an AO algorithm needed to reach these optima under different scenarios. Specifically, to compare the performance of DANTE with other AL algorithms regarding the number of data points required to achieve the global optimum in a quantitative and cost-effective manner, we selected six widely used synthetic functions (known for their difficulty in locating the global optimum) as the validation source, with dimensions ranging from 20 to 2,000 (for example, Ackley, Rastrigin, Rosenbrock, Griewank, Schwefel and Michalewicz functions; Methods, Supplementary Note and Supplementary Table 1). We present and analyze the key results in Table 1 and Extended Data Tables 1 and 2. For example, The Rastrigin function is highly multimodal, featuring numerous local maxima in the ground-truth landscape. The Rosenbrock function contains a long valley with multiple local maxima (Supplementary Fig. 2). These features make these functions ideal benchmarks for assessing the performance of AL algorithms.

We demonstrate that DANTE is most effective when integrated with the DNN. Figure 3d–f shows representative examples (Ackley-100d, Rastrigin-100d and Rosenbrock-60d) comparing the performance of the DNN with six mainstream regression models (that is, decision trees, random forests, linear regression, kernel ridge regression, Gaussian processes and support vector machines). The results indicate that DANTE successfully converges to the global optimum on the Ackley-100d, Rastrigin-100d and Rosenbrock-60d tasks, requiring approximately 500, 2,000 and 5,000 data points, respectively. Notably, DANTE exhibits an exponential convergence rate, quickly approaching near-optimal solutions after 100, 1,500 and 2,000 data points. By contrast, when DANTE is combined with other ML surrogate models,

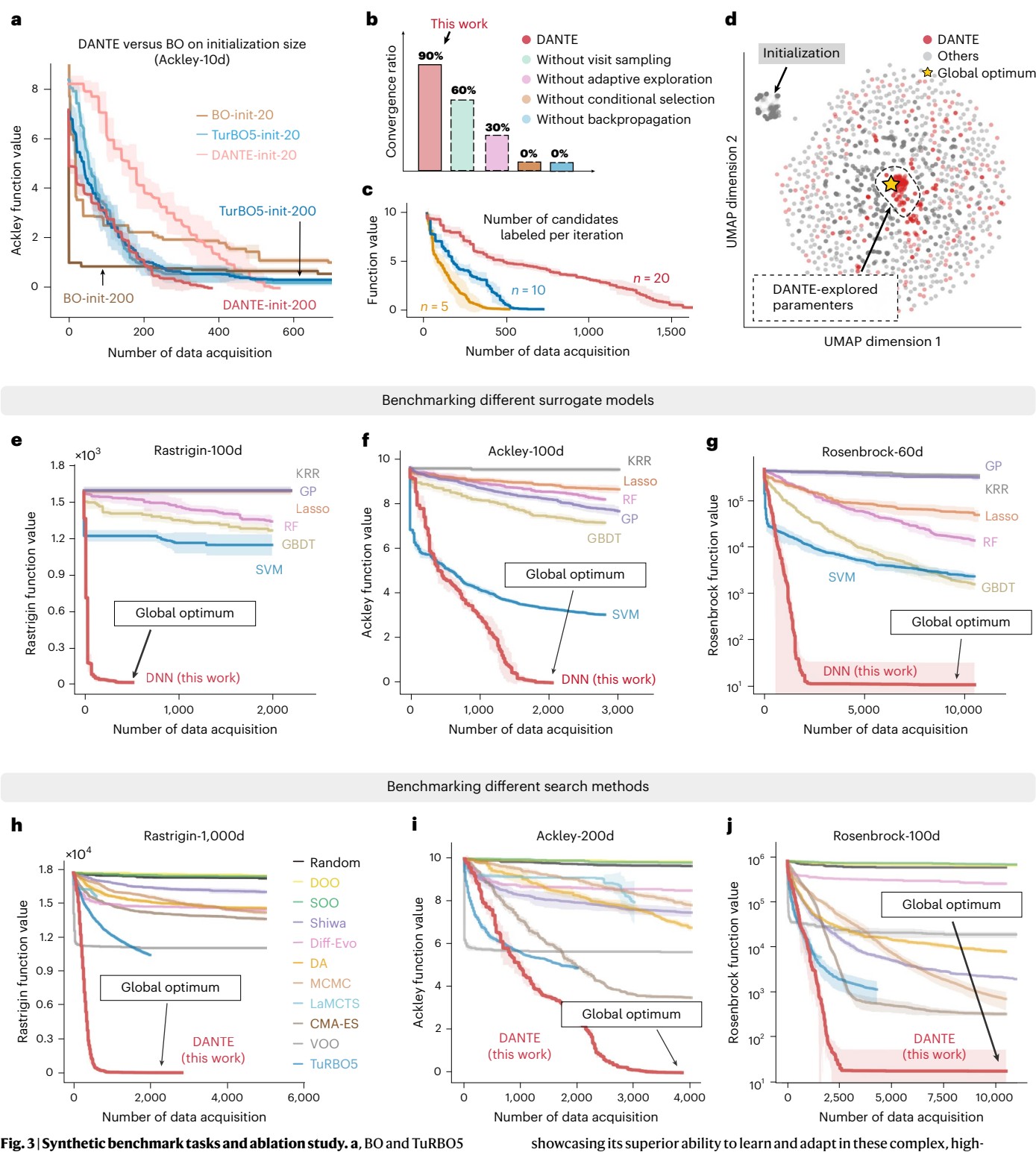

**Fig. 3 | Synthetic benchmark tasks and ablation study. a**, BO and TuRBO5 converge faster when less initial data (~20) are provided, whereas DANTE converges faster to the global optimum when more initial data (~200) are available. Data are presented as mean values ± s.d., $n = 5$. $f(x)$ represents the value of the function. **b**, An ablation study using the Rosenbrock-100d function, evaluated by convergence ratio required to reach the global optimum. $n = 10$. **c**, A smaller sampling batch size leads to a faster convergence rate. Data are presented as mean values ± s.d., $n = 5$. **d**, UMAP visualization of the search trajectories for DANTE and its ablated variants, demonstrating that DANTE efficiently identifies and concentrates on the vicinity of the near-optimal region. **e–g**, The learning progress of DANTE on the Rastrigin-100d (**e**), Ackley-100d (**f**) and Rosenbrock-60d tasks (**g**), highlighting the performance of DANTE equipped with various ML models. DNN consistently outperforms other models,

showcasing its superior ability to learn and adapt in these complex, high-dimensional optimization landscapes. KRR, kernel ridge regression; GP, Gaussian process; Lasso, least absolute shrinkage and selection operator; RF, random forest; GBDT, gradient-boosted decision trees; SVM, support vector machine. Data are presented as mean values ± s.d., $n = 5$. **h–j**, The learning progress of various search methods on the Rastrigin-1,000d (**h**), Ackley-200d (**i**) and Rosenbrock-100d (**j**) functions, highlighting DANTE's fast convergence rate toward the global optimum. DOO, deterministic optimistic optimization; SOO, simultaneous optimistic optimization; VOO, Voronoi optimistic optimization; Diff-Evo; differential evolution; DA, dual annealing; LaMCTS, latent action MCTS; TuRBO, trust region BO; CMA-ES, covariance matrix adaptation evolution strategy. Data are presented as mean values ± s.d., $n = 5$.

**Table 1 | Lowest value achieved by various AL methods on synthetic benchmarks**

| | Ackley-20 | Ackley-100 | Rastrigin-20 | Rastrigin-100 | Rosenbrock-20 | Rosenbrock-100 | Schwefel-20 | Griewank-20 |
|---|---|---|---|---|---|---|---|---|
| Unit | 1 | 1 | $\times 10^2$ | $\times 10^3$ | $\times 10^4$ | $\times 10^4$ | $\times 10^3$ | 1 |
| Maximum number of samples | 1,600 | 2,800 | 1,000 | 2,000 | 6,300 | 10,500 | 1,000 | 1,000 |
| Random | 7.59 ± 0.17 | 9.23 ± 0.13 | 2.18 ± 0.15 | 1.47 ± 0.016 | 2.380 ± 0.119 | 64.60 ± 0.936 | 5.50 ± 0.11 | 233.1 ± 25.49 |
| TuRBO5 | 0.37 ± 0.14 | 1.73 ± 0.18 | 0.52 ± 0.04 | 0.40 ± 0.034 | 0.003 ± 0.000 | 0.127 ± 0.066 | 2.84 ± 0.79 | 1.177 ± 0.049 |
| LaMCTS | 1.96 ± 0.75 | 5.05 ± 0.73 | 0.80 ± 0.30 | 0.82 ± 0.044 | 0.008 ± 0.005 | 0.652 ± 0.098 | 3.32 ± 0.33 | 0.956 ± 0.047 |
| CMS-ES | 0.75 ± 0.09 | 2.85 ± 0.04 | 0.78 ± 0.03 | 0.97 ± 0.017 | 0.006 ± 0.004 | 0.037 ± 0.004 | 5.28 ± 0.44 | 236.7 ± 45.85 |
| Diff-Evo | 6.43 ± 0.16 | 8.13 ± 0.19 | 1.88 ± 0.12 | 1.30 ± 0.032 | 0.797 ± 0.115 | 28.30 ± 2.690 | 5.10 ± 0.17 | 127.6 ± 12.25 |
| DA | **0.00 ± 0.00** | 3.28 ± 0.19 | 1.29 ± 0.06 | 0.53 ± 0.039 | 0.005 ± 0.003 | 0.908 ± 0.088 | 2.38 ± 0.39 | 1.252 ± 0.264 |
| Shiwa | 4.43 ± 0.07 | 5.78 ± 0.52 | 2.48 ± 0.02 | 1.19 ± 0.047 | 2.266 ± 0.146 | 0.240 ± 0.022 | 5.49 ± 0.32 | 0.175 ± 0.246 |
| MCMC | **0.00 ± 0.00** | 4.79 ± 0.16 | 0.89 ± 0.27 | 0.73 ± 0.038 | 0.011 ± 0.006 | 0.088 ± 0.036 | 2.11 ± 0.86 | 5.858 ± 8.782 |
| DOO | 7.17 ± 0.37 | 9.44 ± 0.09 | 2.22 ± 0.14 | 1.50 ± 0.044 | 1.640 ± 0.456 | 72.22 ± 2.700 | 5.56 ± 0.29 | 164.2 ± 21.41 |
| SOO | 7.75 ± 0.18 | 9.40 ± 0.17 | 2.24 ± 0.08 | 1.54 ± 0.027 | 2.760 ± 0.744 | 76.30 ± 2.700 | 2.89 ± 2.18 | 87.67 ± 4.048 |
| VOO | 2.44 ± 0.49 | 5.23 ± 0.17 | 1.03 ± 0.13 | 0.92 ± 0.028 | 0.006 ± 0.000 | 2.107 ± 0.324 | 5.38 ± 0.08 | 0.121 ± 0.091 |
| DANTE | **0.00 ± 0.00** | **0.00 ± 0.00** | **0.00 ± 0.00** | **0.00 ± 0.00** | **0.0003 ± 0.0005** | **0.002 ± 0.004** | **1.20 ± 0.49** | **0.000 ± 0.000** |

Results are averaged over five trials, with ± indicating the s.d. The global optimum for these functions is 0. The bold font denotes the best results in this column.

it often becomes trapped in local optima, remaining notably distant from the global optimum even after utilizing 10,000 data points. These results suggest that the DNN is superior in learning and representing the complexities of the nonlinear search space (our DNN comprises a series of convolutional layers (more than 5), followed by pooling, dropout and normalization layers; for further details, see the Methods, Supplementary Note and Supplementary Fig. 3).

Overall, the evidence presented in Fig. 3a,e suggests that BO performs well in low-dimensional settings, whereas DANTE excels at navigating and locating optima within approximately high-dimensional landscapes. These findings emphasize that selecting the most suitable pairing of surrogate and search models—based on the problem's dimensionality and nonlinearity—is crucial for achieving optimal overall performance.

**From low to high dimensions, from easy to hard data acquisition**

We conduct a thorough ablation study and summarize our results in the Methods. We demonstrate that DANTE consistently outperforms other search methods. Specifically, we evaluate DANTE alongside 11 SOTA algorithms across various categories, including heuristic, Bayesian and tree-based methods. For algorithms lacking a surrogate model, we use DNN as the surrogate. Table 1 and Extended Data Table 1 present benchmark results for the best-achieved values and the number of samples required to reach the global optimum across various synthetic functions, each with a global optimum of 0. The data demonstrate that DANTE consistently attains the global optimum with the fewest data points in most tasks, whereas most competing methods fail to achieve the global optimum altogether. As indicated in Fig. 3a, the BO-based algorithm converges faster than DANTE at low dimensions (<10) and with small initial datasets (<20), while DANTE shows a better performance with higher dimensions and bigger initial datasets. In addition, Fig. 3c shows that a smaller sampling batch size leads to a faster convergence rate. More benchmark results are presented in Supplementary Figs. 4–7.

Figure 3h–j depicts the learning progress of various methods on three high-dimensional tasks: Rastrigin-1,000d, Ackley-200d and Rosenbrock-100d, each tested five times with different random seeds. It is evident that DANTE converges notably faster than all baseline algorithms, while some baselines fail to run due to memory constraints.

Notably, DANTE identifies the global optimum of Rastrigin-1,000d with just 3,000 data points, whereas other baselines struggle with the vast search space, showing minimal progress. A detailed summary of the benchmark results regarding data acquisition for optimal performance is presented in Extended Data Table 1. It is clear that most AL algorithms fail to reach the global optimum for these tasks within the available data limits (for additional results, see Supplementary Figs. 8–10).

We select four noise-free, real-world tasks with relatively easy data access: (1) neural network architecture search on CIFAR-10, aimed at optimizing architecture for maximum test accuracy[34] on the CIFAR-10 dataset[35]; (2) optimization of complex concentrated alloys (CCAs) for improved magnetic properties and resistivity; (3) the optimal control problem of lunar landing, seeking to maximize landing reward; and (4) resolution optimization of transmission electron microscopy (TEM) images. Notably, the search space for these real-world tasks is often constrained by nonlinear external conditions, adding complexity to the learning process and limiting the selection of baseline methods. Further technical details on these benchmarks are provided in the Methods.

Figure 4 demonstrates that DANTE notably outperforms other AL methods across these real-world tasks. While we consider DANTE and RL (for example, policy proximal optimization (PPO)) to pertain to distinct categories of methodologies in terms of (1) quantity of data needed, (2) data accessibility and (3) nature of reward (Fig. 4a), they can still be compared under specific conditions in the lunar landing task, such as a fixed initial position and random seeds. Under these conditions, DANTE demonstrates comparable, or even better, performance compared with PPO, particularly in the initial stages where PPO essentially performs at a random level, indicating its need for a large amount of data (Fig. 4d). However, a notable advantage of PPO is its adaptability, allowing it to be trained for varying environments, such as different initial positions and speeds. In the neural network architecture search task, it achieves near-optimal accuracy of 94.1% with only 200 data points, compared with the global optimum of 94.3%. In the magnetic CCA task, it identifies compositions with 20% higher magnetic properties using just 140 data points. For the lunar landing task, by converting the problem into a noncumulative optimization through fixating the initial positions and predesigned actions at set time intervals (Supplementary Note), DANTE achieves an average reward of 100 after 10,000 samples, whereas other methods remain below 50. In the

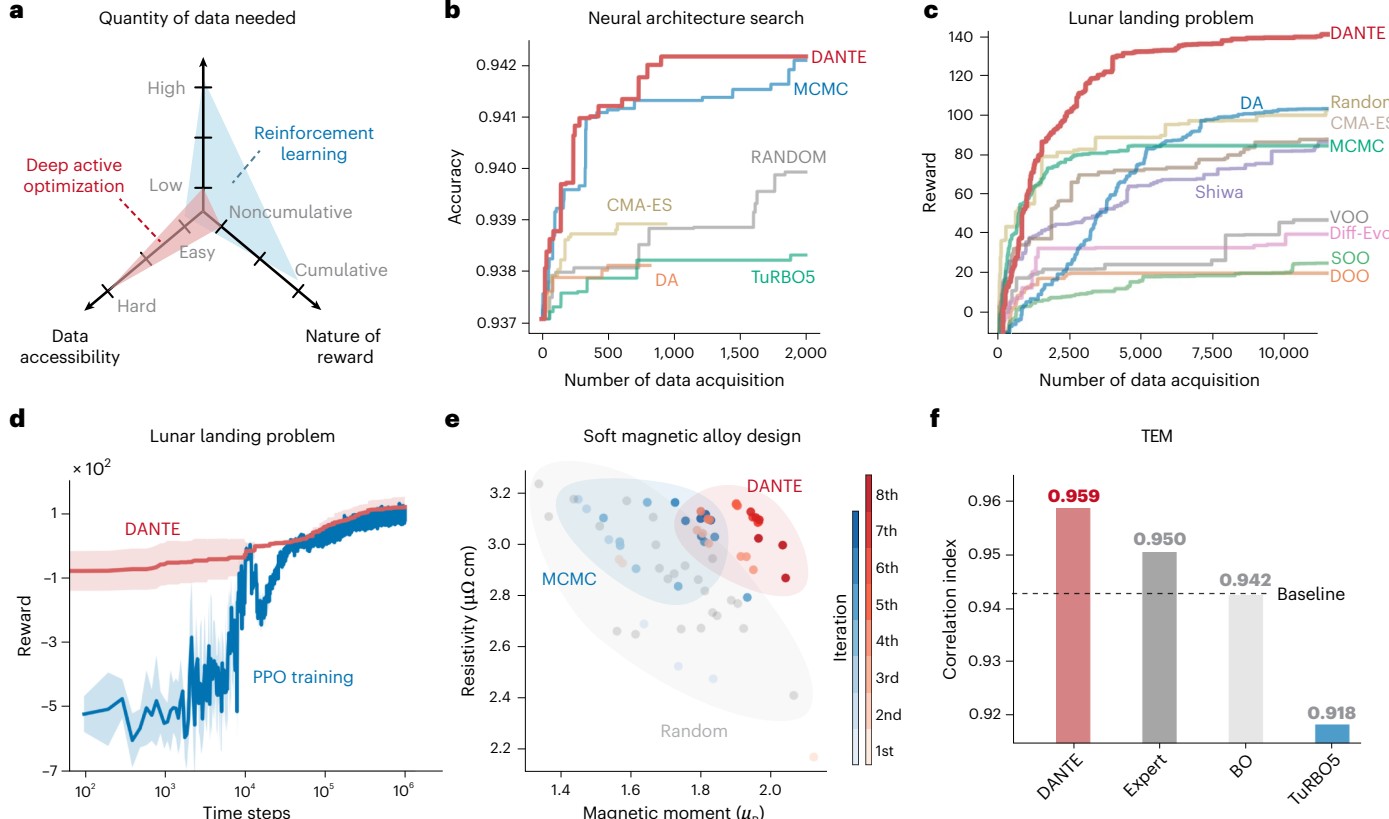

**Fig. 4 | Real-world benchmark tasks. a**, Deep AO is different from RL in terms of quantity of data needed, data accessibility and nature of the reward. **b**, Neural network architecture search on CIFAR-10. Data are presented as mean values, $n = 5$. **c**, The lunar landing problem. Data are presented as mean values, $n = 5$. **d**, In the lunar landing problem, DANTE demonstrates comparable, or even better, performance compared with PPO, particularly in the initial stages where PPO essentially performs at a random level, indicating its need for a large amount of data (fixed random seed). However, a notable advantage of PPO is its adaptability, allowing it to be trained for varying environments, such as different initial positions and speeds. Data are presented as mean values ± s.d., $n = 5$. **e**, Searching for soft magnetic alloy with high resistivity. **f**, Resolution optimization of TEM images is guided by correlation index. DANTE framework outperforms expert's choice, BO and TuRBO5.

TEM resolution optimization task, reconstruction quality is evaluated using a correlation index, which compares the phase of simulated and reconstructed transmission functions (Supplementary Note). DANTE achieves the highest score of 0.958, surpassing even the human expert (details in Supplementary Fig. 11 and Supplementary Table 2).

More real-world problems that involve larger search spaces, more external constraints, noisy labels and highly nonlinear input–output relationships can be found in Extended Data Fig. 1 and Supplementary Note. In these cases, the labels may contain various forms of noise, and acquiring them is both resource intensive and time-consuming, with the optimum often remaining elusive. We demonstrate that the DANTE framework can effectively address these complex tasks without relying on large datasets.

## Discussion

Looking ahead, the current bottleneck lies in the expressive power of the surrogate model and available computer memory rather than in DANTE's inherent capacity. There is potential for DANTE to further push the boundaries of dimensionality by using more sophisticated surrogate models and leveraging larger computing resources, enabling it to address extremely high-dimensional and nonlinear problems beyond 2,000 dimensions in a data-driven manner. We envision numerous opportunities to apply our method across various quantitative sciences. One particularly promising avenue for future application is the integration of our approach with robotic systems to facilitate automated experimental design, thereby accelerating materials discovery and synthesis. Another interesting potential application lies in financial optimization, where the objective is to allocate resources effectively to maximize returns or achieve specific financial goals. We anticipate that our algorithms will soon become standard practice, seamlessly integrated with virtual or experimental setups across multiple disciplines to tackle high-dimensional and nonlinear optimization tasks that were previously deemed intractable. This interdisciplinary approach holds great promise for unlocking further solutions and advancing research and practice in various fields.

## Methods

### Framework of AO

We summarize our key innovations as follows:
- A data-driven formula that leverages the number of visits and ML from a small initial dataset to effectively manage the exploration–exploitation trade-off. This markedly differs from the UCB formula utilized by MCTS, which relies on the average node value and the number of visits derived from numerous simulations.
- Local backpropagation that ensures a balanced exploration–exploitation trade-off for the noncumulative reward problems.
- Adaptive exploration mechanism that favors exploration over exploitation under certain circumstances.
- A modified epsilon-greedy sampling technique that samples best-scored candidates and most-visited candidates at the same time.

While Fig. 1 provides the flowchart illustration of the AL loop, We provide a mathematical formulation framework of the AL problem (referring to materials science as a demonstrator).

Specifically, let $X$ denote the input space (representing, for example, materials such as chemical compositions, specific crystalline structures and so on). Let $Y$ represent the output space, where $y \in Y$ ($y < +\infty$) denotes the specific property or property spectrum of interest (for example, mechanical strength or resistivity) The goal is to identify the optimal material $x^* \in X$ that maximizes or minimizes a property while minimizing the number of labeled data points required. The initial labeled dataset $L$ consists of $D = \{(x_i, y_i)\}_i^n$, where $n$ is the number of initial data points ($n = 200$ in this study). $x$ is the input vector, $X$ is defined as the search space, typically $\mathbb{R}^N$, and $N$ is the dimension. $f$ is the deterministic function that maps the input $x$ to the ground-truth label $y$. The surrogate model $f_\theta$ learns the input-label relation through the dataset $D = \{(x_i, y_i)\}_i^n$, and $n$ is the number of labels and $y_i$ is the label of $x_i$.

The AL loop involves iteratively selecting the samples from $X$, based on a search algorithm $Q(x; f_\theta)$, and retraining the surrogate model $f_\theta$. At each iteration $t$:

(1) Model training: train the model $f_\theta$ using the current labeled dataset $D$:

$$\theta^t = \mathrm{argmin}_\theta \mathbb{E}_{(x,y)D}[L(\theta; x, y)],$$

where $L$ is the loss function.

(2) Search and selection: select a subset of $k$ samples $x_{\text{new}} \in X$ based on $f_\theta$ using a search algorithm $Q(x; f_\theta)$ ($k = 20$ in both benchmark and real-world studies):

$$x_{\text{new}} = \mathrm{argmax}_{x \in X} f_\theta.$$

(3) Labeling and updating: obtain the labels $y_{\text{new}}$ for the selected samples $x_{\text{new}}$, and add them to

$$D \leftarrow D \cup \{(x_{\text{new}}, y_{\text{new}})\}.$$

RL is another commonly used method for identifying optimal solutions. Differences in AL and RL lie in three main aspects: (1) data accessibility, (2) the quantity of data needed and (3) the nature of rewards (noncumulative versus cumulative).

- Data accessibility: In typical RL settings, a policy network interacts with the environment, requiring easy access to reward functions. By contrast, AO, particularly in scientific contexts, often deals with limited access to reward functions. For instance, in materials science, it might take months to obtain just a few labeled data points.
- Quantity of data needed: RL training commonly demands large amounts of labeled data or observations to develop an effective policy network. AL, however, operates in a low-data regime, usually with fewer than 1,000 data points, and requires only a value-estimation network.
- Nature of rewards: RL algorithms are primarily used for trajectory planning and optimal control problems, involving sequential decisions and cumulative rewards. Conversely, AL typically focuses on maximizing the current reward functions.

### Technical details of NTE
There are three modes of action for the stochastic expansion, each occurring with equal probability (that is, 1/3). (1) One-step move: this mode represents the smallest possible change at a single position of the feature vector. (2) Single mutation: in this mode, one position of the feature vector randomly mutates to any value within the allowed range. (3) Scaled random mutation: this mode involves a proportion of the feature vector randomly mutating to any allowed values. The number of leaf nodes equals the dimension of the feature vector.

A real-world complex system often can be represented as a vector or a matrix. For example, in materials science, searching for a high-performance CCAs can be formulated as optimizing the properties by tuning the alloy compositions[36]; In biology, the protein design can be approached as improving biofunctionalities by optimizing a sequence consisting of 20 amino acids. We implement a convolutional neural network for the deep learning surrogate model. It consists of convolutional layers and is followed by pooling, dropout and normalization layers to prevent overfitting. The network parameters are optimized using Adam Optimizer, and the loss function is the mean-squared error or mean absolute percentage error. More detailed parameters are found in the Supplementary Note and Supplementary Fig. 3.

Standard MCTS consists of four major steps: selection, expansion, simulation and backpropagation (Supplementary Fig. 12). We summarize key differences between DANTE and MCTS.

(1) The MCTS backpropagation mechanism uses the result of the rollout to update both value and visitation of the nodes along the path, which affects all nodes (from root to end node) at a global level. The local backpropagation updates only the visitation information of the current root node and the subsequent leaf nodes. We do not update the value information because our optimization problem focuses solely on discovering a single optimal state and retains little 'memory' of previous states. Therefore, value information is not backpropagated, and visitation backpropagation is short-ranged, relying only on nearby visitation data to guide exploration.
(2) MCTS selection step chooses the leaf node with max UCB and proceeds to the next expansion with the selected leaf node, whereas the expansion of NTE is conditioned on an inequality of the DUCB: the expansion proceeds with the leaf node that has a higher DUCB than root node; otherwise, it proceeds with the same root.
(3) Conventional rollout uses the simulation step to reach the end state (for example, win or loss of a game) and uses the average value as the current node value, while the stochastic rollout of NTE does not need the simulation step to obtain the node value; instead, it uses the surrogate model to estimate the node value.

Furthermore, Supplementary Fig. 13 shows the difference between UCB and DUCB.

### Top-visit sampling
The sampling technique for selecting top candidates is a critical component of the AO pipeline. An effective sampling method should identify the most informative candidates while preserving data diversity, ensuring the surrogate model generalizes well to unseen data. The widely used sampling approach is the epsilon-greedy method, which combines greedy selection with random sampling. To enhance the generalization capability of the surrogate model, we extend the epsilon-greedy strategy by implementing 'top-visit sampling', which samples data that are frequently visited during rollouts. Figure 3b demonstrates that DANTE, when lacking top-visit sampling, exhibits a higher surrogate model loss and requires 30% more data points to achieve the global optimum (as detailed in the Methods, Supplementary Note and Supplementary Fig. 1).

### Ablation study
We conducted an ablation study on the Rosenbrock-100d function to analyze the impact of DANTE's individual components on overall performance (Fig. 3b; additional results are provided in Supplementary Fig. 1). The results clearly show that conditional selection and local backpropagation are critical to DANTE's effectiveness. Without conditional selection, the tree search suffers from the value deterioration problem and has a 0% convergence ratio, defined as the frequency with which the algorithm identifies the global optimum within

a given number of data points. Without local backpropagation, DANTE becomes a greedy stochastic tree search, leading to poor performance and similarly a 0% convergence ratio. Moreover, omitting top-visit sampling and adaptive exploration, while still allowing for convergence, notably degrades performance. In such cases, the average best $f(x)$ remains distant from the global optimum, and the convergence ratios drop to 60% and 30%, respectively.

We further assess the limits of DANTE by evaluating its performance in tackling high-dimensional problems with a limited number of data points. As shown in Extended Data Table 2, DANTE demonstrates exceptional performance, successfully converging across various synthetic functions ranging from 200 to 2,000 dimensions. By contrast, SOTA methods fail to converge to any functions beyond 100 dimensions. Notably, none of the baseline methods achieves global convergence on the Rosenbrock function in dimensions exceeding 10, while DANTE successfully converges in dimensions as high as 200 (Supplementary Table 3).

## Synthetic functions

The synthetic functions are designed for evaluating and analyzing the computational optimization approaches. In total, six of them are selected on the basis of their physical properties and shapes. Results for the Ackley, Rosenbrock and Rastrigin functions are presented in the main text because they are widely studied and relevant results are extensively available in the literature. We also test three other synthetic functions (Griewank, Schwefel and Michalewicz), and the results are presented in Supplementary Fig. 10. The Ackley function can be written as

$$f(x) = -a \times \exp\left(-b\sqrt{\frac{1}{d}\sum_{i=1}^{d}x_i^2} - \exp\left(\frac{1}{d}\sum_{i=1}^{d}\cos(cx_i)\right) + a + \exp(1),\right. \quad (2)$$

where $a = 20$, $b = 0.2$, $c = 2\pi$ and $d$ is the dimension.

The Rosenbrock function can be written as

$$f(x) = \sum_{i=1}^{d-1}\left[100(x_{i+1} - x_i^2)^2 + (x_i - 1)^2\right]. \quad (3)$$

The Rastrigin function can be written as

$$f(x) = 10d + \sum_{i=1}^{d-1}\left[x_i^2 - 10\cos(2\pi x_i)\right]. \quad (4)$$

The three functions are evaluated on the hypercube $x_i \in [-5, 5]$, for all $i = 1, \ldots, d$ with a discrete search space of a step size of 0.1; we also show that different step sizes (within a certain range) do not affect the general behavior of the algorithm (Supplementary Fig. 14). We sample 20 data points per round when using neural networks as surrogate models. More details and results can be found in the Supplementary Note.

## Electron ptychography

**Feature engineering.** The feature vector consists of eight variables: beam energy, defocus, maximum number of iterations, number of iterations with identical slices, probe-forming semi-angle, update step size, slice thickness and number of slices. Detailed values and their bounds are listed in Supplementary Table 4.

**Optimization target.** The objective function NMSE is calculated between the positive square root of the measured diffraction pattern $I_M$ and the modulus of the Fourier-transformed simulated exit-wave $\Psi$, which can be formulated as

$$\frac{1}{N}\sum_{i}^{N}\left|\sqrt{I_{M(i)}(\mathbf{u})} - |\mathcal{F}[\Psi_i(\mathbf{r})]|\right|^2, \quad (5)$$

where $\mathbf{r}$ and $\mathbf{u}$ denote the real- and reciprocal-space coordinate vectors, respectively, and $N$ is the total number of the measured diffraction patterns.

**Correlation index.** The degree of matching for a given template $T$ by intensity function $P$ is characterized by a correlation index, which can be defined by the following relation:

$$\frac{\sum_{i=1}^{m}P(x_i, y_i)T(x_i, y_i)}{\sqrt{\sum_{i=1}^{m}P^2(x_i, y_i)}\sqrt{\sum_{i=1}^{m}T^2(x_i, y_i)}}, \quad (6)$$

where $(x_i, y_i)$ is the coordinate of pixel $i$.

**Dataset simulation.** abTEM[37], an open-source package, is used for the simulation of a TEM experiment. For this case study, we simulated a four-dimensional dataset of 18-nm-thick silicon along the [110] direction with Poisson noise.

**Ptychographic reconstruction.** The analysis is performed using py4DSTEM[38], a versatile open-source package for different modes of STEM data analysis. See Supplementary Figs. 11 and 15 for more details about the reconstruction process.

## Architected materials

**Feature engineering.** In this study, the objective for architected materials optimization is a Gyroid triply periodic minimal surface structure, which naturally occurs in butterfly wings and is renowned for its exceptional biological characteristics and mechanical performance. The Gyroid scaffold to be optimized comprises 27 subunits with a dimension of $2 \times 2 \times 2$ mm, allowing for tuning its geometry features and mechanical properties by adjusting each subunit's density. The density of each subunit can take discrete values from 10% to 80%, with an increment of 10%. The base material of the scaffold is Ti6Al4V alloy. Three-dimensional convolutional neural networks are used to accurately and rapidly assess the impact of the adjustments of the subunit's density on the scaffold's performance. Details about structure generation are presented in ref. 39.

**Optimization target.** To mechanically stimulate bone reconstruction in bone defects, it is well recognized that the elastic modulus of bone grafts should be equivalent to that of the replaced bone, which ranges from 0.03 to 3 GPa for cancellous bone and 3 to 30 GPa for cortical bone, while there are specific modulus demands for different anatomical locations[40]. Moreover, it requires the optimization of load-bearing capacity to prevent damage during implantation. Here, we establish the modulus requirement for the implanted site at 2.5 GPa. Consequently, the optimization target is to maximize the yield strength of the scaffold while ensuring the elastic modulus remains within a specified range (2,500 ± 200 MPa).

**Finite element simulation.** Finite element (FE) simulations of the compressive stress–strain curves of scaffolds are conducted using ABAQUS 2018. The FE simulations utilize the same rigid-cylinder and deformable-implant-structure model. The material property is set to be homogeneous with a Poisson's ratio of 0.25; more details in the calibration protocol were developed in ref. 39. Ductile damage is used to simulate plastic deformation up to the failure stage, with a fracture strain set at 0.03. The effects of triaxiality deviation and strain rate are disregarded. Displacement and force are extracted during postprocessing and subsequently converted to strain and stress, respectively. FE simulation agrees well with the experiment compression curves (Supplementary Fig. 16).

**ML model.** The initial dataset (100 density matrices) is consistent with our previous work[39], and the corresponding elastic modulus and yield strength are calculated by FE simulations. Three-dimensional

convolutional neural networks are used to predict the elastic modulus and yield strength of the scaffolds with varying density matrices. The model architecture comprises an input layer, convolutional layers, fully connected layers and an output layer (refer to the Supplementary Note and Supplementary Fig. 17 for detailed parameters). In the input layer, the scaffold structure is voxelized into $60 \times 60 \times 60$ pixels, where each pixel denotes either the solid phase (1) or void phase (0) within the scaffold. The convolutional layers are designed with a series of three-dimensional convolution kernels to extract high-dimension information about the scaffold, while the output layer delivers the final prediction.

### Compositionally complex alloys

**Feature engineering.** We adopt 27 elements: Fe, Co, Ni, Ta, Al, Ti, Nb, Ge, Au, Pd, Zn, Ga, Mo, Cu, Pt, Sn, Cr, Mn, Mg, Si, Ru, Rh, Hf, W, Re, Ir and Bi, to design six-element CCAs with either bcc or fcc structures. For Fe, Co and Ni, the atomic ratio ranges from 0 at.% to 100 at.%, while for other elements, it ranges from 0 at.% to 40 at.%, with 0.5 at.% intervals. In addition, the total atomic percentage of Fe, Co and Ni is designed to fall between 60 at.% to 80 at.%. For CCAs with a bcc crystal structure, the Fe/(Co + Ni) ratio is required to be greater than or equal to 1.5, whereas for fcc structures, it is required to be less than or equal to 1.5.

**Optimization target for magnetic and electric properties.** The optimization target is to maximize the following target:

$$Target = M \times \rho, \tag{7}$$

where $M$ stands for magnetic moment and $\rho$ for resistivity.

**Optimization target for transport properties.** The optimization target is to maximize the following target:

$$Target = AHC \times AHA, \tag{8}$$

while keeping the formation energy under the upper limit of 0.02.

**Density functional calculation.** The transport properties are described by the conductivity tensor $\sigma_{\nu\mu}$ ($\nu, \mu = x, y, z$). The anomalous Hall conductivity (AHC, $\sigma_{xy}$) and anomalous Hall angle (AHA, $\sigma_{xy}/\sigma_{xx}$) are determined in the frame of Kubo–Bastin linear response formalism within relativistic multiple-scattering Korringa–Kohn–Rostoker (KKR) Green's function (GF) method[41], which has been implemented in the MUNICH SPR-KKR package[42]. The Kubo–Bastin formalism includes both the Fermi-surface and Fermi-sea contributions to equal footing, in which the Fermi-surface term contains only contribution from states at the Fermi energy ($E_F$) while the Fermi-sea term involves all the occupied states (with energy $E$) below the Fermi energy, that is,

$$\sigma_{\mu\nu} = \sigma_{\mu\nu}^I + \sigma_{\mu\nu}^{II} \tag{9}$$

$$\sigma_{\mu\nu}^I = \frac{\hbar}{2\pi\Omega} Tr \left\langle \hat{j}_\mu \left( \hat{G}^+ - \hat{G}^- \right) \hat{j}_\nu \hat{G}^- - \hat{j}_\mu \hat{G}^+ \hat{j}_\nu \left( \hat{G}^+ - \hat{G}^- \right) \right\rangle \tag{10}$$

$$\sigma_{\mu\nu}^{II} = \frac{\hbar}{2\pi\Omega} \int_{-\infty}^{E_F} Tr \left\langle \hat{j}_\mu \hat{G}^+ \hat{j}_\nu \frac{d\hat{G}^+}{dE} - \hat{j}_\mu \frac{d\hat{G}^+}{dE} \hat{j}_\nu \hat{G}^+ \right.$$
$$\left. - \left( \hat{j}_\mu \hat{G}^- \hat{j}_\nu \frac{d\hat{G}^-}{dE} - \hat{j}_\mu \frac{d\hat{G}^-}{dE} \hat{j}_\nu \hat{G}^- \right) \right\rangle. \tag{11}$$

The electric current operator is given by $\hat{j}_{\mu(\nu)} = -|e|c\alpha$, with $e > 0$ being the elementary charge. $\hat{G}^+$ and $\hat{G}^-$ denote the retarded and advanced GFs, respectively. The representation of the GFs for the first-principles treatment of equations (10) and (11) leads to a product expression containing matrix elements of the current operators with the basis

functions and $k$-space integrals over scattering path operators. In this averaging procedure, the chemical disorder and vertex corrections are treated by means of coherent potential approximation[43]. For both Fermi surface and surface terms, the conductivity tensor partitions into an on-site term $\sigma^0$ involving regular and irregular solutions and an off-site term $\sigma^1$ containing only regular solutions. This formalism has been validated to provide consistent residual and anomalous Hall resistivities with experiments[41]; more details can be found in the Supplementary Note.

**ML model.** Initial 200 CCAs are randomly generated following the previously described design rules, and their corresponding AHA, AHC and formation energy are calculated by density functional theory (DFT). For CCAs with bcc grain structures, 154 configurations ultimately converge in the DFT calculations, whereas for fcc structures, there are 178. We train one-dimensional convolutional neural networks to predict the AHA, AHC and formation energy of the CCAs. The model architecture includes an input layer, convolutional layers, fully connected layers and an output layer (see the Supplementary Note and Supplementary Fig. 18 for detailed parameters).

### Cyclic peptide binder

**Feature engineering.** We represent the cyclic peptide as a sequence of integers that range from 0 to 19, with each number corresponding to a distinct type of canonical amino acid. The leaf node within the DANTE framework is obtained through stochastic expansion. In this process, two complementary strategies are used: one that introduces random mutations in existing sequences and another that generates entirely new sequences, ensuring a comprehensive exploration of the sequence space.

**Optimization target.** The optimization target of cyclic peptide binder is defined as follows:

$$Target = SC \times dSASA/100, \tag{12}$$

where SC stands for shape complementarity, and dSASA represents the change in solvent-accessible surface area before and after interface formation. The SC value ranges from 0 to 1, referring to how well the surfaces of two proteins fit geometrically together at their interface; dSASA measures the size of the interface (in units of $Å^2$). Both metrics are essential to assess the quality of the interface. Therefore, we multiply these two metrics to formulate a multiobjective optimization problem, which is used to evaluate the performance of DANTE.

**Dataset.** Fourteen unique protein and canonical cyclic peptide complexes are sourced from the Protein Data Bank, with peptide lengths ranging from 7 to 14 amino acids. We perform three different optimization tasks using DANTE, gradient descent (GD) and Markov chain Monte Carlo (MCMC). The tasks start from a random initial sequence. The structure with the highest target value is selected as the best structure. For each task, we performed three independent tests.

**Alphafold2 settings.** The structure of protein and cyclic peptide binder complex is predicted by Alphafold2-multimer implemented in ColabDesign. A modified offset matrix for the relative positional encoding of a target protein and cyclic peptide complex is adapted to give the structure with high accuracy[44]. For designing a cyclic peptide binder, the binder hallucination protocol is utilized for both GD and MCMC methods. In this study, we maintain the length of the cyclic binder and the interaction site hotspots consistent with those found in nature. For GD, the method 'design_pssm_semigreedy()' is used, setting soft_iter to 120 and hard_iter to 32. The loss function is a weighted sum of pLDDT (predicted local distance difference test) and interface contact loss, with other parameters left at their default settings. For

the MCMC method, a total of 1,000 steps are executed to find the sequence achieving the highest pLDDT. More detail can be found in the Supplementary Note.

**Rosetta interface analyzer.** The SC and dSASA values for the predicted structure of the protein and cyclic peptide complex are computed using the Rosetta Interface Analyzer. Initially, the Rosetta minimize protocol is applied to obtain the structure with minimum energy proximal to the initial conformation. To ensure that cyclic peptides within the complex retain their cyclic nature and do not become linear, the options '-use_truncated_termini' and '-relax:bb_move false' are used. Subsequently, the minimized complex serves as the input for the interface analyzer.

## Data availability
Source data are provided with this paper. All initial datasets in this work are randomly generated. Source data are also available via GitHub at https://github.com/Bop2000/DANTE/, as well as via Zenodo at https://doi.org/10.5281/zenodo.16225698 (ref. 45).

## Code availability
Code for DANTE is available at via GitHub at https://github.com/Bop2000/DANTE/, as well as via Zenodo at https://doi.org/10.5281/zenodo.16225698 (ref. 45).

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

## Acknowledgements

This work is funded by Tsinghua-Toyota Joint Research Fund; National Natural Science Foundation of China (grant number 52175274); CityUHK start-up fund (grant number 9382006); National Natural Science Foundation of China (grant number 52301302); and Beijing Natural Science Foundation (grant number L244002). We acknowledge the computing time provided to them at the NHR Center NHR4CES at RWTH Aachen University (project number p0024007). This is funded by the Federal Ministry of Education and Research and the state governments participating on the basis of the resolutions of the Gemeinsame Wissenschafts Konferenz for national high-performance computing at universities (www.nhr-verein.de/unsere-partner).

## Author contributions

Y.W. conceived the idea; Y.W. and B.P. developed the theory and methods. B.P. and Y.W. implemented the algorithms. B.P., Y.W., R.X., P.-Y.T., Y.Q., Y.C. and S.B. carried out the numerical studies and analysis; R.X. performed the DFT calculations; Y.Q. performed the FE methods analysis; Y.C. built the cyclic peptide design pipeline; P.-Y.T. developed the electron ptychography simulation pipeline; P.-Y.T., B.P. and Y.W. produced the final figures; Y.W. and B.P. wrote the original draft; all authors contributed to data analysis, discussions and manuscript preparation.

## Funding

## Competing interests

The authors declare no competing interests.

## Additional information

**Supplementary information Extended data** is available for this paper at https://doi.org/10.1038/s43588-025-00858-x.

**Correspondence and requests for materials** should be addressed to Ye Wei or Dierk Raabe.

**Extended Data Table 1 | Number of data points needed to achieve the global optimum**

| | Ackley-20 | Ackley-100 | Rastrigin-20 | Rastrigin-100 | Rosenbrock-20 | Rosenbrock-100 | Schwefel-20 | Griewank-20 |
|---|---|---|---|---|---|---|---|---|
| **Max # of samples** | 1,600 | 2,800 | 1,000 | 2,000 | 6,300 | 10,500 | 1,000 | 1,000 |
| **Random** | N.A. | N.A. | N.A. | N.A. | N.A. | N.A. | N.A. | N.A. |
| **TuRBO5** | N.A. | N.A. | N.A. | N.A. | N.A. | N.A. | N.A. | N.A. |
| **LaMCTS** | N.A. | N.A. | N.A. | N.A. | N.A. | N.A. | N.A. | N.A. |
| **CMS-ES** | N.A. | N.A. | N.A. | N.A. | N.A. | N.A. | N.A. | N.A. |
| **Diff-Evo** | N.A. | N.A. | N.A. | N.A. | N.A. | N.A. | N.A. | N.A. |
| **DA** | 428 ± 114 | N.A. | N.A. | N.A. | N.A. | N.A. | N.A. | N.A. |
| **Shiwa** | N.A. | N.A. | N.A. | N.A. | N.A. | N.A. | N.A. | N.A. |
| **MCMC** | 408 ± 65 | N.A. | N.A. | N.A. | N.A. | N.A. | N.A. | N.A. |
| **DOO** | N.A. | N.A. | N.A. | N.A. | N.A. | N.A. | N.A. | N.A. |
| **SOO** | N.A. | N.A. | N.A. | N.A. | N.A. | N.A. | N.A. | N.A. |
| **VOO** | N.A. | N.A. | N.A. | N.A. | N.A. | N.A. | N.A. | N.A. |
| **DANTE** | **292 ± 65** | **1788 ± 139** | **220 ± 74** | **444 ± 55** | **4263 ± 1111** | **3098 ± 1031** | N.A. | **860 ± 115** |

Results are averaged over 5 trials, and ± denotes the standard deviation. N.A. (Not Available) indicates that it does not reach the global optimum.

**Extended Data Table 2 | The max convergence dimensions on synthetic functions**

|  | Ackley | Rastrigin | Rosenbrock | Griewank |
|---|---|---|---|---|
| DANTE | 1500 | 2000 | 200 | 500 |
| SOTA | 100 | 100 | 10 | 60 |

This table shows the highest dimension at which DANTE and SOTA methods (Random search, DOO, SOO, VOO, Shiwa, differential evolution, dual annealing, MCMC, LAMCTS, CMA-ES, TuRBO5) achieve global convergence, with DANTE outperforming SOTA on higher-dimensional tasks.

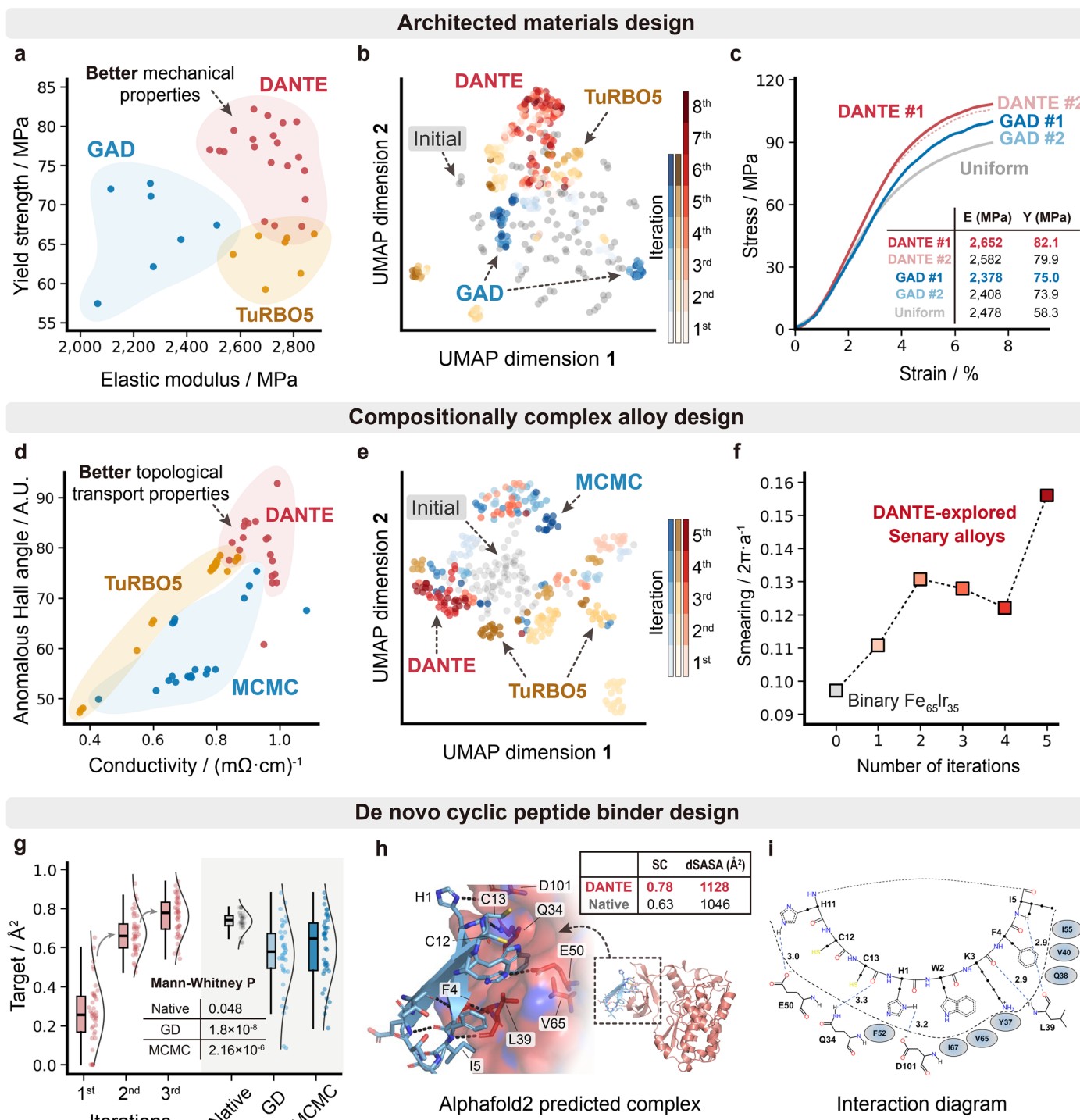

**Extended Data Fig. 1 | High-dimensional, high-cost real-world tasks.**
(**a**) Optimizing the mechanical properties of architected materials by DANTE and generative architecture design (GAD, baseline). (**b**) Uniform manifold approximation and projection (U-MAP) 2D representation of results from two methods. (**c**) Stimulated strain-stress curve of both methods. The inlet shows the density matrix. (**d**) Optimizing the electronic properties of complex concentrated alloys (CCAs) by DANTE and MCMC (baseline). (**e**) U-MAP 2D representation of input distribution from both methods. (**f**) The curves along a selected momentum path on the Fermi surface, a quantitative measure for describing the smearing. (**g**) Optimizing the protein-protein interactions (PPIs) using DANTE and other two methods. Box plots indicate median (middle line), 25th, 75th percentile (box), and 1.5 × interquartile range (whiskers). $n = 41$ for 1st and 2nd iterations, $n = 42$ for 3rd iteration, Native, GD (gradient descent), and MCMC. (**h**) An example of Alphafold2 predicted complex (pdbid: 4ib5). The cyclic peptide is designed by DANTE. (**i**) Interaction diagram of DANTE peptide with the target protein.

