## [Peer Review File · Nature Computational Science]

Deep active optimization for complex systems

Corresponding Author: Dr Ye Wei

A version of this paper was originally rejected for publication by Nature Computational Science, however that decision was reconsidered after appeal by the authors.

Version 0:

Decision Letter:

Dear Dr Wei,

Your manuscript "Deep active learning for complex systems" has now been seen by 4 referees, whose comments appear below. In the light of their advice, we have decided that we cannot offer to publish your manuscript in Nature Computational Science.

From the reports, you will see that while they find your work of some potential interest, the referees raise concerns about the advance your findings represent over earlier work and the strength of the novel conclusions that can be drawn at this stage. In particular, referees raised critical concerns, such as the lack of enough comparisons to important algorithms, the limited novelty in your algorithm, the inconsistency in dimensionality between synthetic and practical examples, etc.

Indeed, it seems that a significant amount of additional work and clarification would be necessary in order to address the aforementioned points, and it is not clear whether or not your present results will remain strong in light of this additional work. Therefore, we feel that these criticisms are sufficiently important as to preclude publication of your work in Nature Computational Science.

Should future experimental data allow you to address the following points, we would be willing to look at a revised manuscript and potentially reconsider the work at our discretion (unless, of course, something similar has by then been accepted at Nature Computational Science or appeared elsewhere). This includes submission or publication of a portion of this work somewhere else.

- Clarify the novelty of your framework (including providing more benchmarks)
- Provide quantitative comparisons as requested by our referees (such as the comparison to reinforcement learning, etc.) to better demonstrate the advance of your method
- Provide real-world examples that have similar dimensionality as that in the synthetic benchmarks to better demonstrate the practical value

If you are interested in submitting a suitably revised manuscript in the future or if you have any questions, please contact me. In case you are not interested in submitting a revised version, you may transfer your manuscript to another journal in the Nature Portfolio using the link I provide at the end of this email.

I am sorry that we cannot be more positive on this occasion, but hope that you find the referees' comments helpful when preparing your paper for resubmission elsewhere.

Best regards,

Jie Pan, Ph.D.
Senior Editor
Nature Computational Science

Reviewers' Comments:

Reviewer #1 (Remarks to the Author):

The manuscript “Deep Active Learning for Complex Systems“ describes an algorithmic approach for sample-efficient global optimization of high-dimensional black-box problems. The proposed strategy consists of training a deep neural network surrogate on all available observations. In the second step, a modified version of Monte-Carlo Tree Search is used to recommend new candidates for experimental evaluation, with the aim of finding the x that maximizes the surrogate model’s predictions. The authors systematically benchmark their algorithm on very high-dimensional analytical surfaces, where it significantly outperforms several alternative surrogate models or acquisition strategies. Moreover, case studies on different scientific discovery problems are presented.

I come from the application of global optimization algorithms to scientific problems, and my review will therefore be less focused on detailed mathematical formulation or implementation. I fully agree with the authors that the optimization of high-dimensional black-box functions is crucial for science and engineering. However, based on the presentation in the article, I am unsure where the main advance of the work lies. Training a surrogate model is a standard workflow in iterative optimization. Therefore, the key novelty must lie in the adapted tree search method to find the x in X for which the surrogate model m predicts the highest value. This effectively replaces the “acquisition function optimization” used in standard low-dimensional active learning workflows (e.g. Bayesian Optimization). In BO terms, the presented optimization algorithm could be viewed as optimizing a fully greedy acquisition function.

Especially when targeting applications in the natural sciences, the paper needs a clearer discussion of when or why this approach is advantageous. The general discussion remains vague: is it particularly useful in the low-data regime, in cases of high nonlinearity, or primarily for high-dimensional problems? Why not use a (BO-like) gradient-based optimization of the DNN surrogate?

In this context, I find especially Fig. 3a and the related discussion somewhat misleading. If sample efficiency is a main concern (in relatively low-dimensional spaces), is there evidence that deep active learning is superior? I would argue that for low-dimensional, low-data problems, classical BO (e.g. using Gaussian Process surrogates) can be more sample efficient, because the surrogate model can provide good predictivity with as few as 10–20 examples. In this work, the authors always initialize their model with 200 observations, which is rather high for many real-world experimental settings. Although I appreciate the idea of the plot, the figure and the text must be much clearer about in which regimes the DAL actually constitutes an advantage.

A second main concern is that the workflow does not address the exploration–exploitation tradeoff when optimizing the unknown objective function (similar to a “fully greedy acquisition function optimization”, see above). In classical active learning, this is often proposed to be crucial for enabling sample-efficient optimization. Is this a problem for experimental use cases? In the manuscript, this possible limitation is hidden behind a very detailed discussion of the algorithm that is used for optimizing the “acquisition function” (i.e. the DNN predictions) over the full search space. Would a combination with cheap uncertainty estimates for DNN models (e.g. Laplace approximation or variational inference) be feasible to circumvent this?

Further comments and questions:

- DNN architecture: How is it chosen? Is generalizable to different problem types, or are problem-specific models needed?
- Metrics: The authors primary evaluate the number of experiments required to find the global optimum. Is this truly the most relevant metric? For many scientific applications, identifying the single best optimum is less important than discovering several top solutions.
- Dimensionality of synthetic and experimental benchmarks: The authors demonstrate success on high-dimensional benchmark surfaces, but the “real-life” scientific examples have significantly lower dimensionality. How would BO-type methods perform in these cases?
- Peptide example: The featurization is based on one-hot encoding, so the distances do not reflect any physical meaning. Some of the expansion policies become unclear in this context.
- Batch-wise evaluation: In their experiments, the authors show few iterations with batch sizes of 20 experiments per iteration. What is the computational cost of generating each batch of recommendations? Would a smaller batch size yield improved sample efficiency?

Reviewer #2 (Remarks to the Author):

This manuscript uses a deep active learning method that combines deep neural networks with a tree search to accelerate the search for optimal solution in various problems. The results seem useful, however, there are some issues need to be addressed. The main concern is about the novelty of the strategy and its usability in practical case. Thus, I cannot recommend it for publication in the present form.

1. The authors claim that, different from the MCTS method used in examples like AlphaGo, they propose a method incorporates novel mechanisms such as UCB. This is not fair, it is known that UCB is a popular utility function and has been used in many optimization problem. The author should clarify what is the novelty in method compared to previous ones.
2. One of the key issues the manuscript plans to address is the high-dimensional space, for example, the one with 2000 dimensions. They demonstrate their method in several scenarios. However, it is questionable if this is the real case in practical applications. At least in materials science, the dimension is often not as high as 2000. More importantly, for high

dimensions feature engineering is usually first used to eliminate the features with poor information, which can help simplify the surrogate model, increase the interpretability, and save computational resource. So it is difficult to imagine why the authors want to apply the method to such a high dimensional space.

3. One page 3, "Traditional machine learning models, such as Bayesian methods, heavily rely on assumptions about prior distributions or feature engineering". This statement is too strong, there are many machine learning models, not all the models have assumptions on data distribution etc.

4. The organization of the manuscript is too wordy, e.g., the introduction of AL on page 3 and page 7.

5. The authors first compare different surrogate models and find DNN is the fastest, then different selectors are compared and the DA outperforms the others. Can this process ensure that DNN+DA is the best, as not all the possible combinations are examined.

Reviewer #2 (Remarks on code availability):

The codes are organized well and the README file is in detail.

Reviewer #3 (Remarks to the Author):

The authors propose Deep Active Learning with Neural-Surrogate-Guided Tree Exploration (DANTE). This method is as an optimization algorithm for arbitrary objective functions.

DANTE combines value estimation through a Deep Neural Network (DNN) with a tree search algorithm to explore the solution space. The tree search is performed using a modified version of the Upper Confidence Bound (UCB) algorithm termed Data-driven Upper Confidence Bound (DUCB). The difference between UCB and DUCB is that the formula contains various modifications. Other changes in the tree exploration mechanism are stochastic rollout (imparting random changes on the selected state to create new leaf nodes), conditional selection (reverting back to the root node if it achieves a higher DUCB value compared to all its leaves), and local backpropagation (the visitation counts are only updated for the root and the leaf node instead of the entire path). These modifications, together with the DUCB algorithm, conform the method termed Neural-surrogate-guided Tree Exploration (NTE). The greedy-epsilon technique is used to select the new samples among the top candidates with a modification to favor the selection of the most-visited samples during stochastic rollout.

DANTE starts by training the value-estimation network on available data. Since this method is intended to work in fields where data collection is expensive, the tree search with NTE selects favorable samples which are then labeled. Then, the DNN is trained again including the new data and the process continues until a desirable solution is found. It must be noted that, in DANTE, unlike typical active learning settings, the selection score for the samples is the same objective function to be optimized.

The authors benchmark DANTE against several optimization methods in a wide variety of test cases, including non-convex optimization, materials design, (non-cumulative) lunar landing, and cyclic peptide binder design. I commend the authors for testing their algorithm in problems spanning such a wide array of fields. These tests show that DANTE performs better than other tested algorithms.

I believe that DANTE is a promising method for optimization in high-dimensional systems, but I would like the authors to attempt some comparison to reinforcement learning methods.

Comments

1. My main comment is related to the classification of DANTE as an active learning method. I would suggest that the authors revise this decision upon deeper review of the reinforcement learning literature. From the presentation of the algorithm, it seems that the only connection to active learning is that a few promising samples are selected for the training of the surrogate model after each iteration. Active learning typically has the goal of reducing the error or uncertainty in the trained model while labeling the least number of data possible. There are many examples available in the literature, [1] and [2] are just some recently published ones. In the unnumbered formula on line 146, page 8, the authors seem to interpret the active learning selection criterion as simply maximizing the value from the surrogate model, rather than making any attempts to quantify uncertainty, effect of the sample on the model, or other metric to select maximally informative samples. On the other hand, this understanding aligns very well with the reward-maximization goal of reinforcement learning. Of course, there are many variations in machine learning methods and the differences between one framework and the other may blur. Nonetheless, DANTE draws upon decades of reinforcement learning developments (tree search, UCB, greedy-epsilon, value-estimation networks, AlphaGo) and almost nothing on active learning theory.

2. The issue of presenting the method as an active learning technique is not merely a semantic one. The lack of discussion of reinforcement learning also means that comparisons against other methods may conveniently omit some techniques. In particular, in line 323, page 19, the authors write "DANTE significantly outperforms other AL methods" but there seems to be no effort to compare against reinforcement learning techniques. For example, why was soft actor-critic [3] not used in comparison for the lunar landing benchmark where it is possible to frame the problem as a cumulative reward objective? Wouldn't an algorithm that performs sequential decision making be more suitable in that case?

3. In order to revise the manuscript, I think the authors need to provide a deeper discussion of reinforcement learning works so that readers can compare DANTE with other methods that apply similar ideas. It might be helpful to explain the

distinguishing features between active and reinforcement learning in the introduction to clarify why the authors place the method in one category or the other. Finally, I think that a comparison to at least one modern reinforcement learning method must be provided for at least one of the benchmarks.

References

- [1] Kulichenko, Maksim, et al. "Uncertainty-driven dynamics for active learning of interatomic potentials." *Nature Computational Science* 3.3 (2023): 230-239.
- [2] Yang, Jason, et al. "Active learning-assisted directed evolution." *Nature Communications* 16.1 (2025): 714.
- [3] Haarnoja, Tuomas, et al. "Soft actor-critic: Off-policy maximum entropy deep reinforcement learning with a stochastic actor." *International conference on machine learning*. Pmlr, 2018.

Reviewer #3 (Remarks on code availability):

I tested the code from the GitHub on the simple case presented in the README file (Ackley-20d) and the code ran correctly (found the optimal solution) without errors.

Reviewer #4 (Remarks to the Author):

In the manuscript "Deep active learning for complex systems", a novel deep active learning method is proposed to address optimization problems characterized by high dimensionality, limited data availability, and non-cumulative objectives. The method integrates a surrogate deep neural network with a new tree-exploration search strategy, significantly reducing data requirements while effectively navigating high-dimensional search spaces. By introducing novel components in the tree-exploration search module, the approach successfully tackles both real-world and synthetic challenging problems, outperforming state-of-the-art optimization methods, including Bayesian optimization and evolutionary algorithms.

The work is technically sound and presents a novel contribution. The proposed method has been rigorously evaluated against most state-of-the-art optimization algorithms on both toy problems and complex real-world challenges. The reviewer recommends publication after addressing a few concerns.

The approach has been applied to real-world problems, including architected materials design, where the objective function involves maximizing a structure's yield strength while maintaining a constant Young's modulus. This suggests that DANTE can also handle multi-objective optimization problems. The authors are encouraged to discuss this aspect explicitly in the manuscript and provide additional multi-objective case studies (in the Supplementary). Moreover, are there limitations regarding the number of objectives the method can handle? Can it effectively discover the Pareto front in such cases? While the manuscript is well-written, the section "Neural-Surrogate-Guided Tree Exploration" is convoluted and lacks clarity. Improving the flow of this section, as well as the overall readability of the manuscript, would enhance comprehension. Additionally, in Eq. (1), the variables N and n are not introduced or explained. Clarifying their definitions would improve the mathematical clarity of the paper.

Reviewer #4 (Remarks on code availability):

The code includes clear installation instructions in the README file. The reviewer appreciates its clarity and usability.

Although we cannot publish your paper, it may be appropriate for another journal in the Nature Portfolio. If you wish to explore the journals and transfer your manuscript please use our manuscript transfer portal. You will not have to re-supply manuscript metadata and files, unless you wish to make modifications. For more information, please see our [manuscript transfer FAQ](http://www.nature.com/authors/author_resources/transfer_manuscripts.html?WT.mc_id=EMI_NPG_1511_AUTHORTRANSF&WT.ec_id=AUTHOR) page.

For Nature Portfolio general information and news for authors, see <http://npg.nature.com/authors>

Version 1:

Decision Letter:

** Please ensure you delete the link to your author homepage in this e-mail if you wish to forward it to your co-authors. **

Dear Dr Wei,

Your manuscript "Deep active learning for complex systems" has now been seen by 4 referees, whose comments are appended below. You will see that while they find your work of interest, they have raised points that need to be addressed

before we can make a decision on publication.

The referees' reports seem to be quite clear. Naturally, we will need you to address *all* of the points raised.

While we ask you to address all of the points raised, the following points need to be substantially worked on:

- Please provide the comparisons to other BO surrogate models as requested by referee #1
- Please discuss the insights on what element makes your model better than others
- Please respond to referee #2's comments, e.g., the influence of descriptor dimension to optimization efficiency, the influence of descriptor numbers on recovery capability.

You will also need to make some editorial changes so that it complies with our Guide to Authors at <https://www.nature.com/natcomputsci/for-authors>.

In particular, I would like to highlight the following points of our style:

Nature Computational Science titles should give a sense of the main new findings of a manuscript, and should not contain punctuation. Please keep in mind that we strongly discourage active verbs in titles, and that they should ideally fit within 150 characters each (including spaces).

To improve the accessibility of your paper to readers from other research areas, please pay particular attention to the wording of the paper's abstract, which serves both as an introduction and as a brief, non-technical summary in about 150 words. It should include the background and context of the work, 'Here we show' or an equivalent phrase, and then the major results and conclusions of the paper. Because researchers from other sub-disciplines will be interested in your results and their implications, it is important to explain essential but specialised terms concisely. We suggest you show your summary paragraph to colleagues in other fields to uncover any problematic concepts.

We encourage you to archive the data reported in your manuscript in an accessible, persistent repository. If your data are archived prior to the acceptance of your manuscript, please provide us with the full citation as soon as you receive it so that a link to the data can be included in the publication. See <http://www.nature.com/authors/policies/availability.html> for more information.

If your paper is accepted for publication, we will edit your display items electronically so they conform to our house style and will reproduce clearly in print. If necessary, we will re-size figures to fit single or double column width. If your figures contain several parts, the parts should form a neat rectangle when assembled. Choosing the right electronic format at this stage will speed up the processing of your paper and give the best possible results in print. If you are in doubt about the correct format for your figures after reading our guidelines, please ask the art editors for advice computationalscience@nature.com.

Figure legends must provide a brief description of the figure and the symbols used, including definitions of any error bars employed in the figures.

As a guideline, Articles allow up to 50 references (excluding those cited exclusively in Methods).

Please include a statement before the Acknowledgements naming the author to whom correspondence and requests for materials should be addressed.

Finally, we require authors to include a statement of their individual contributions to the paper -- such as experimental work, project planning, data analysis, etc. -- immediately after the acknowledgements. The statement should be short, and refer to authors by their initials. For details please see the Authorship section of our joint Editorial policies at <http://www.nature.com/authors/policies/authorship.html>.

Please use the following link to submit your revised manuscript and a point-by-point response to the referees' comments (which should be in a separate document to any cover letter):

Link Redacted

** This url links to your confidential homepage and associated information about manuscripts you may have submitted or be reviewing for us. If you wish to forward this e-mail to co-authors, please delete this link to your homepage first. **

To aid in the review process, we would appreciate it if you could also provide a copy of your manuscript files that indicates your revisions by making use of Track Changes or similar mark-up tools. Please also ensure that all correspondence is marked with your Nature Computational Science reference number in the subject line.

In addition, please make sure to upload a Word Document or LaTeX version of your text, to assist us in the editorial stage.

If you have any issues when updating your Code Ocean capsule during the revision process, please email the Code Ocean support team Cc'ing me.

To improve transparency in authorship, we request that all authors identified as 'corresponding author' on published papers create and link their Open Researcher and Contributor Identifier (ORCID) with their account on the Manuscript Tracking

System (MTS), prior to acceptance. ORCID helps the scientific community achieve unambiguous attribution of all scholarly contributions. You can create and link your ORCID from the home page of the MTS by clicking on 'Modify my Springer Nature account'. For more information please visit www.springernature.com/orcid.

We hope to receive your revised paper within three weeks. If you cannot send it within this time, please let us know.

Best regards,

Jie Pan, Ph.D.
Senior Editor
Nature Computational Science

Reviewers comments:

Reviewer #1 (Remarks to the Author):

The authors have made remarkable efforts to revise their manuscript, and I highly appreciate their extensive responses (even though at times, it was difficult to distill the key aspects due to the overwhelming amount of information). As part of their revision, the authors did a number of important benchmark experiments to support the superior optimization performance of DANTE compared to established optimization algorithms.

Additionally, I thank the authors for clarifying my misinterpretation regarding the role of the tree search algorithm. I found the statement that it manages the exploration–exploitation tradeoff in a frequentist manner very convincing, and I feel that the clarity of the manuscript could benefit from a similar explanation, too.

That being said, I have one concern regarding the impact of DANTE that has not been sufficiently addressed, which is related to the comparison with Bayesian Optimization.

- In all discussions and experiments shown in the paper, the authors appear to limit BO to Gaussian Process surrogate models (even though this is not stated explicitly). This is somewhat misleading, since BO can employ any Bayesian surrogate model, including Bayesian Neural Networks.

- That said, a key question arises: What specifically makes DANTE superior to BO in the presented experiments? Is it the fact that the DNN surrogate model is more expressive compared to a GP? Or are the performance gains mainly attributed to the tree search algorithm to effectively navigate the candidate space. In my opinion, the latter would be significantly more impactful, and would require a key experiment involving a BO workflow that uses the exact same DNN model as DANTE as the surrogate. Approximate Bayesian inference methods (e.g. Laplace approximations or last-layer variational inference) should be readily available for neural networks. From this experiment, the contribution of the tree search method should be estimated. The response letter and the Supplementary Materials briefly mention a “CNN-BO” experiment, but due to insufficient details provided, it is difficult to assess precisely what has been done. This experiment, however, would be crucial for understanding the method’s novelty and its primary contribution.

Minor comments and remarks:

- I agree with Reviewer #3 that the term “Active Learning” (also used in the title) should be used with caution. Typically, active learning refers to iterative workflows that aim at selecting data points that improve the model’s predictivity, which is not the goal of this work.

- I appreciate the authors’ clarification regarding evaluation metrics. I fully agree that the number of experiments or function evaluations should be the primary metric. However, my initial comment was rather about emphasizing the practical relevance of scoring the top-k candidates rather than solely the top-1 candidate.

- Especially in the introduction, the use of the term “dataset” is somewhat confusing, especially given that the data continually changes in an iterative workflow. To improve clarity, clear distinctions should be made between initially available data and newly acquired data throughout the iterations.

Reviewer #1 (Remarks on code availability):

At the first glance, the repository looks reasonably organised. A README file is provided that provides clear instructions on how to use the code. However, I have not attempted to install and use the code myself.

Reviewer #2 (Remarks to the Author):

The authors partially addressed my concerns. Especially for the high dimension problem, they presented two examples to show that high dimension exists. However, this is not general for real cases in materials science or alloy design. For example, the ice structure optimization, it is not a feasible way to use as many descriptors as possible as inputs, in particular to the materials or physics scientists. In general, the design of descriptors, such as order parameters based on expertise, is important in such studies and no body will want to collect or use too many descriptors, which complicates the problem concerned and makes the surrogate model or optimization a whole black-box. Using descriptors that are not quite important

seems like "Grasp the shadow and lose the substance". If such idea works, then for materials or alloys design we can easily generate descriptors more than 1000, but I am not sure why we should do this. Also, for the implant design, is there some rules such as periodicity that can be used to simplify the optimization? The dimensions should not merely rely on the size of implant itself that cannot be simplified, just like the structure optimization of component in aerospace, which is even complex than the case here.

The authors also argue that more descriptors can improve the recovery capability. This may be correct, but there can be only a little decrease in recovery when many unimportant descriptors are removed. Just like the influence of descriptors dimension on the accuracy of surrogate model, more descriptors sometimes cannot increase but decrease the accuracy.

In addition, I am wondering that for a certain problem, how the dimension change of descriptor affects the optimization efficiency? Because the dimension might change the distribution of the search space.

The authors also states that the descriptors from feature engineering is often suitable to a specific property. I cannot fully agree with this, as many cases have shown that if the descriptors can be properly defined based on, for example, domain knowledge or theory, they can generalize to other properties well. This is underpinned by the robust generalizability of theory or knowledge, for example, the Hall-Petch equation.

Thus, I cannot be convinced by the present response. I still question on why authors want to focus on the high dimension optimization and the results cannot fairly support their conclusion.

Reviewer #2 (Remarks on code availability):

The codes are organized well.

Reviewer #3 (Remarks to the Author):

Authors have successfully addressed the comments from the first round of review.

Reviewer #3 (Remarks on code availability):

Assessment was provided in the first round of review. There are not any major changes made in the code.

Reviewer #4 (Remarks to the Author):

All concerns have been fully addressed. I recommend acceptance.

Reviewer #4 (Remarks on code availability):

Documentation and installation instructions are clear, and the code for reproducing results is provided. I did not attempt to run it.

Version 2:

Decision Letter:

Our ref: NATCOMPUTSCI-24-2514B

11th June 2025

Dear Dr. Wei,

Thank you for submitting your revised manuscript "Deep active learning for complex systems" (NATCOMPUTSCI-24-2514B). It has now been seen by the original referees and their comments are below. The reviewers find that the paper has improved in revision, and therefore we'll be happy in principle to publish it in Nature Computational Science, pending minor revisions to satisfy the referees' final requests and to comply with our editorial and formatting guidelines.

TRANSPARENT PEER REVIEW

Nature Computational Science offers a transparent peer review option for original research manuscripts. We encourage increased transparency in peer review by publishing the reviewer comments, author rebuttal letters and editorial decision letters if the authors agree. Such peer review material is made available as a supplementary peer review file. **Please remember to choose, using the manuscript system, whether or not you want to participate in transparent peer review.**

Thank you again for your interest in Nature Computational Science. Please do not hesitate to contact me if you have any questions.

Sincerely,

Jie Pan, Ph.D.
Senior Editor
Nature Computational Science

ORCID

Reviewer #1 (Remarks to the Author):

The authors have convincingly addressed my remaining concerns. In particular, they should be commended for the detailed comparison with conventional BO, which clearly demonstrates that DANTE's superior performance stems from the combination of the DNN surrogate and the MCTS-type recommendation system. This finding is very intriguing and, in my view, impactful for practical applications. That said, I am happy to recommend the paper for publication.

However, the authors appear to have ignored my earlier concern regarding the use of the term "active learning", which was also expressed by other reviewers in previous rounds of review. In the field, "active learning" typically refers to iterative workflows aimed at sample-efficient data acquisition to improve model predictivity. In contrast, the approach in this work focuses on sample-efficient data acquisition for optimizing an unknown black-box function, which resembles the classical use case of BO. While these differences are subtle, they have a number of practical implications. Therefore, in my opinion, it is important to make this distinction, in order to avoid potential misconceptions about the scope and objectives of the paper.

Version 3:

Decision Letter:

Dear Dr Wei,

We are pleased to inform you that your Article "Deep active optimization for complex systems" has now been accepted for publication in Nature Computational Science.

Once your manuscript is typeset, you will receive an email with a link to choose the appropriate publishing options for your paper and our Author Services team will be in touch regarding any additional information that may be required.

Authors may need to take specific actions to achieve compliance with funder and institutional open access mandates. If your research is supported by a funder that requires immediate open access (e.g. according to [Plan S principles](https://www.springernature.com/gp/open-science/plan-s-compliance) or the [NIH public access policy](https://www.springernature.com/gp/open-science/us-federal-agency-compliance)) then you should select the gold OA route, and we will direct you to the compliant route where possible. Because authors warrant under our subscription licensing terms that they haven't committed to licensing any version of their article under a licence inconsistent with the terms of our agreement – including the applicable embargo period – publication under the subscription model isn't suitable for authors whose funders require no embargo.

Acceptance of your manuscript is conditional on all authors' agreement with our publication policies (see <https://www.nature.com/natcomputsci/for-authors>). In particular your manuscript must not be published elsewhere and there must be no announcement of the work to any media outlet until the publication date (the day on which it is uploaded onto our web site).

Before your manuscript is typeset, we will edit the text to ensure it is intelligible to our wide readership and conforms to house style. We look particularly carefully at the titles of all papers to ensure that they are relatively brief and understandable.

Once your manuscript is typeset, you will receive a link to your electronic proof via email with a request to make any corrections within 48 hours. If, when you receive your proof, you cannot meet this deadline, please inform us at rjsproduction@springernature.com immediately.

If you have queries at any point during the production process then please contact the production team at rjsproduction@springernature.com.

We welcome the submission of potential cover material (including a short caption of around 40 words) related to your manuscript; suggestions should be sent to Nature Computational Science as electronic files (the image should be 300 dpi at 210 x 297 mm in either TIFF or JPEG format). We also welcome suggestions for the Hero Image, which appears at the top of our [home page](http://www.nature.com/natcomputsci); these should be 72 dpi at 1400 x 400 pixels in JPEG format. Please note that such pictures should be selected more for their aesthetic appeal than for their scientific content, and that colour images work better than black and white or grayscale images. Please do not try to design a cover with the Nature Computational Science logo etc., and please do not submit composites of images related to your work. I am sure you will understand that we cannot make any promise as to whether any of your suggestions might be selected for the cover of the journal.

Best regards,

Jie Pan, Ph.D.
Senior Editor
Nature Computational Science

P.S. Click on the following link if you would like to recommend Nature Computational Science to your librarian: <https://www.springernature.com/gp/librarians/recommend-to-your-library>

** Visit the Springer Nature Editorial and Publishing website at <http://editorial-jobs.springernature.com> for more information about our career opportunities. If you have any questions please click [here](mailto:editorial.publishing.jobs@springernature.com).**

Response to the Reviewers' Report

We would like to cordially thank the editor and reviewers for their positive feedback and constructive criticism, which have enabled us to significantly enhance the quality of this manuscript (NATCOMPUTSCI-24-2514). In response to their requests, we have provided a detailed point-by-point response letter, including additional evidence and real-world tasks. Our response is structured as follows. The original comments from the reviewers are copied below in black and italic font. For each comment, we present a detailed response (blue font), a complete description of all the additional experiments and simulations we conducted, and the corresponding manuscript modifications (red font). In the manuscript, the amended parts are shown in red font.

Again, we would like to sincerely acknowledge the editor and reviewers for their positive feedback and constructive criticism, which have significantly enhanced the quality of this manuscript. In response to their requests, we have provided a detailed point-by-point response letter, including additional evidence and real-world tasks. Here below is a summary of the rebuttal letter:

- 1) **Revision:** Revised manuscript designed to improve the readability and clarity of the main novelties. This effort focuses on the rewriting and reorganizing the introduction and result sections. This revision includes 1) Summary of the novelty at the beginning of the result section; 2) different subsections for different components; 3) Additional comparative plots in the Methods sections; 4) Extensive discussion on the reinforcement learning aspects, highlighting the distinction items between reinforcement learning and deep active learning.
- 2) **Clarification:** We fully recognize that DANTE may sometimes be confused with other methods, since it contains features from active learning, reinforcement learning and derivative-free optimization. To clarify, we now elaborate in more detail on how the DANTE approach differs from other methods in the Methods section.
- 3) **Additional benchmarks and Bayesian optimization (BO) baselines:** We introduce additional results and evidence from more benchmark studies with different sampling rate and initial data points. Also, we performed extensive Bayesian baseline calculations and included the results into the existing real-world tasks.
- 4) **Additional high-dimensional real-world tasks:** We have added now one more real-world task (~100 dimensions) - targeting receptor binding domain of SARS-COVID-2 spike protein with Monobody using AlphaFold 3 to further illustrate the practical value of our method (see Figure R9).
- 5) **Remarks on the significance of our method:** We also take the opportunity to clarify that the value of our method has now been demonstrated across nine different real-world tasks spanning the fields of physics, computer science, and biology. The practical significance, however, lies not only in its capability to cope with large dimensionality spaces but also in addressing non-linearity, noise levels, and external constraints associated with these tasks (evidenced in Figure R5). It is important to note that tackling high-dimensional real-world tasks (examples provided in the revised version of the manuscript), particularly those

exceeding 1000 dimensions, poses significant challenges at this time due to the extremely high computational costs, which scale exponentially with dimensionality. We appreciate your understanding of the immense importance of current computational limitations in that context and we are deeply convinced that methods relieving this enormous burden are worth to be pursued, to make the use of AI methods more efficient and sustainable.

Reviewers' Comments:

Reviewer #1 (Remarks to the Author):

Reviewer #1

'However, based on the presentation in the article, I am unsure where the main advance of the work lies. Training a surrogate model is a standard workflow in iterative optimization. Therefore, the key novelty must lie in the adapted tree search method to find the x in X for which the surrogate model m predicts the highest value. This effectively replaces the "acquisition function optimization" used in standard low-dimensional active learning workflows (e.g. Bayesian Optimization).'

Response:

We would like to express our gratitude to the reviewer for their valuable suggestions. In summary, DANTE is a versatile, data-driven optimization algorithm that integrates machine learning techniques with search history to identify optimal solutions for complex problems while utilizing minimal data points.

To further illustrate the advantages of DANTE and its main differences compared to prior approaches, we summarize our key innovation items from two perspectives: state-of-the-art (SOTA) performance and algorithmic novelty.

SOTA Performance: We propose a data-driven algorithm that shows state-of-the-art empirical benchmark results and is able to effectively address real-world problems using minimal data points. This is evidenced by our extensive benchmark study and real-world examples (Main text Table 1, 2 and 3; Main text Figure 3,4 and 5), to further validate our advantage, we **1) add Bayesian baselines to the real-world problems as well as synthetic benchmark with various initial dataset and sampling rates (Figure R1, R2, R3, R4, R5, R7 and R10)**, the results indicate that DANTE outperforms SOTA methods by a large margin. Specifically, we draw the following conclusions:

- At a very small initial dataset (<20) and very low dimension (<10), BO-based methods demonstrate a faster convergence rate but DANTE still achieves a better optima (Figure R2 and R3) if sufficient data points are sampled.

Figure R1. Bayesian optimization converges faster than DANTE at low dimension (~ 10) and small number of initial dataset (~ 20).

Figure R2. Selected benchmark performance at initial dataset = 10. Test functions are Ackley-10d, 20d; Rastrigin-10d,20d; Rosenbrock-10, 20d. Selected algorithms: BO, TurBO5 (State-of-the-art BO-variant), BO-CNN and DANTE. It can be observed that the BO-based method can often have a faster convergence rate at the beginning. But eventually, as data accumulates, DANTE is able to find a better optima. Notably, as dimensionality increases, DANTE demonstrates clearer advantages.

Figure R3. Selected benchmark performance at initial dataset =20. Test functions are Ackley-10d, 20d; Rastrigin-10d, 20d; Rosenbrock- 10, 20d. Selected algorithms: BO, TurBO5 (State-of-the-art BO-variant), BO-CNN and DANTE. It can be observed that the BO-based method can often have a faster convergence rate at the beginning. But eventually, as data accumulates, DANTE is able to find a better optima. As the number of dimensions increases, DANTE demonstrates its advantages more clearly.

- We observe that DANTE converges faster (needs fewer data points) at smaller sampling batch size (Figure R4).

Figure R4. Benchmark performance at different sampling batch size = 5, 10, 20. Test functions are Ackley-10d, 20d; Rastrigin-10d,20d; Rosenbrock- 10, 20d; it can be observed that the smaller sampling batch size leads to a faster convergence rate.

We present now also additional results from the BO-based method, demonstrating that it consistently falls short of achieving superior outcomes when applied to complex and high dimensional real-world tasks characterized by a high level of non-linearity, search constraints, and noise (Figure R5).

Figure R5. Due to factors such as high non-linearity, various constraints, and noise, as well as external limitations, the solution provided by TurBO5, a state-of-the-art Bayesian Optimization (BO) variant, is inferior to that of DANTE.

We argue that there is a considerable distinction between (deep) active learning and reinforcement learning concerning their applicable scopes, even though they are inherently different, they can be evaluated under particular conditions, including a fixed initial position and consistent random seeds. Therefore, we include the reinforcement learning baseline for comparative analysis.

In such scenarios, DANTE exhibits a performance that is comparable to, or sometimes surpasses, that of policy proximal optimization (PPO), especially during the initial phases where PPO tends to operate randomly. Nonetheless, a significant advantage of PPO is its flexibility, which enables it to be trained across diverse environments, such as varying initial positions and speeds. Conversely, any modifications in the environment require the retraining or re-implementation of DANTE. For comprehensive details regarding the experimental setups, please consult the main text.

Figure R6. Comparison of DAL and RL in terms of quantity of data needed, data accessibility and nature of rewards.

Figure R7. Performance comparison (X-axis in normal timestep scale and log timestep scale) of DANTE and PPO under fixed position and random seed.

We contend that the practical significance of an optimization problem is contingent upon its complexity, which is shaped by multiple factors, notably by 1) Dimensionality, 2) Nonlinearity, and 3) The costs associated with the validation source.

The complexity of an optimization problem is influenced not only by its dimensionality but also by its inherent nonlinearity. A general guideline for assessing this complexity can be articulated as follows:

Complexity \propto Dimension \times degree of nonlinearity

For instance, in synthetic tasks, a low-dimensional problem characterized by a highly nonlinear function (e.g., the Schwefel function in 20 dimensions) can present challenges comparable to those encountered in a high-dimensional problem exhibiting relatively low nonlinearity (e.g., the Ackley function in 2000 dimensions). This equivalence arises from the necessity for a similar number of data points to attain the global optimum in both cases.

In real-world applications, the optimization task is frequently subject to various constraints and often contains varying levels of noise, contributing to a heightened degree of nonlinearity (also demonstrated in Figure R5). In this context, we assert that we have effectively demonstrated the practical value of the DANTE algorithm across eight distinct real-world tasks spanning the fields of physics, computer science, and biology.

Secondly, a significant challenge arises from the fact that as dimensionality increases, the computational cost escalates exponentially (Figure R8). Consequently, current hardware limitations make it exceedingly difficult to conduct active learning on real-world problems with 1000 dimensions. This difficulty can be attributed to the following relationship:

Computational cost \propto exp(Dimension)

Thus, the primary bottleneck for tackling high-dimensional real-world problems with AI methods lies in their associated computational costs, which are exceptionally high. However, we assert that such problems do exist and would garner significant interest should hardware improvements occur.

Figure R8. A real-world case study involving bone implant design illustrates that costs increase exponentially with the dimensionality of the design parameters.

Table R2. The comparison of computational costs across varying dimensions of the bone implant problem reveals that both the time required for computation and the number of elements involved escalate dramatically as the dimensionality increases.

Scaffold	Dimension	Compression strain (%)	Number of elements	Time budget (min)
3*3*3	27	6	112085	10.8
5*5*5	125	6	473374	67.2
Rabbit bone	210	6	2019673	1545.6

Real-world task 4: Designing Monobody against SARS-COVID-2

Recent advancements in protein structure prediction software, such as AlphaFold 3, have led to substantial improvement in computational efficiency. This progress enables us to introduce a pertinent high-dimensional example of active learning in the context of protein design, which involves optimizing hundreds of amino acids. As achieving this optimization using AlphaFold 2, as utilized in our current project, presents considerable challenges. We present an example involving hundreds of dimensions, with AlphaFold 3 serving as the validation source. In our

analysis, we also tested several baseline methods, including TurBO (BO variant), MCMC, and DANTE.

Figure R9. Pipeline of designing monobody binder to SARS-COVID-2 spike protein. Search strategies are DANTE, MCMC and TurBO5.

Monoclonal antibodies (mAbs) are among the most common therapeutics and are frequently used for the treatment of cancer, autoimmune diseases, infectious diseases, and more. Recently, monobodies (Mobs) have gained attention as an alternative to mAbs due to their strong developability^{1,2}. Mobs are proteins roughly 100 amino acid residues in length, featuring an immunoglobulin (Ig)-like fold composed of 7 beta strands, derived from human Fibronectin type III (FN3). These proteins combine the advantageous features of both mAbs and miniproteins, which are generated through de novo design approaches, while also overcoming some of their disadvantages.

We build the Monobody binder design pipeline using DANTE, MCMC and TurBO5 targeting at the receptor binding domain (RBD) of SARS-CoV-2 Spike protein. Similar to the real-world tasks proposed in the real-world task 3 in the main text, the pipeline uses AlphaFold 3, Rosetta and ProteinMPNN to generate pools of monobody binders with high ipTM score.

ipTM is a metric used to assess the accuracy of predicted relative positions in protein-protein complexes³. Values greater than 0.8 indicate confident high-quality predictions, suggesting that the model has likely captured the correct orientation and interaction of the subunits. Conversely, values below 0.6 suggest a likely failed prediction, indicating potential inaccuracies in the model. ipTM values falling between 0.6 and 0.8 are considered a grey zone, where predictions could either be correct or erroneous, requiring further validation or additional evidence to ascertain their reliability.

The ipTM distributions obtained from different methods—DANTE, TurBO5, and MCMC—reveal that DANTE significantly outperforms TurBO5 and generates a greater number of high-quality designs compared to MCMC. This indicates that DANTE is more effective at predicting the

accurate relative positions of subunits in protein-protein complexes, leading to more reliable and successful design outcomes.

Figure R10. The ipTM distributions obtained from different methods—DANTE, TurBO5, and MCMC—reveal that DANTE significantly outperforms TurBO5 and generates a greater number of high-quality designs compared to MCMC.

Table R3. Percentile of high-quality solutions designed by three methods.

Methods	Percentile of high-quality solutions (ipTM>0.8)
DANTE	27%
MCMC	7%
TurBO5	0%

Algorithmic novelty

DANTE is a hybrid methodology that synthesizes features from three distinct optimization genres, enriched by key modifications that result in a unique and innovative variant.

We summarize our key innovations as follows:

- **A data-driven formula that leverages the number of visits and ML from a small dataset to effectively manage the exploration-exploitation trade-off.** This markedly differs from the UCB formula utilized by MCTS, which relies on the average node value and the number of visits derived from a large number of simulations.
- **Local backpropagation** that ensures a balanced exploration-exploitation trade-off for the non-cumulative reward problems and accelerates the search for maximal non-cumulative reward.

- **Adaptive exploration mechanism** that favors exploration over exploitation under certain circumstances.
- **A modified epsilon-greedy sampling technique** that samples best-scored candidates and most-visited candidates in one batch.

As a hybrid methodology, we recognize that it may sometimes be confused with other methods. To clarify, we elaborate on how DANTE differs from traditional methods as follows:

1. **Difference from Derivative-Free Optimization Methods:** While our tree search method can be classified as a derivative-free optimization technique, it fundamentally diverges from traditional derivative-free black-box optimization algorithms. Typically, these algorithms are evaluated based on the number of function evaluations. In contrast, the DANTE algorithm utilizes deep neural networks (DNN) as a guiding mechanism and is assessed based on the data points required to achieve optimal results. This approach introduces significant complexity, as it necessitates a closed-loop learning process that requires synergy between surrogate modeling and the sampling strategy.
2. **Difference from Bayesian Methods:** DANTE is inherently a frequentist’s approach and employs the number of visits to facilitate the exploration-exploitation trade-off. Unlike traditional Bayesian black-box optimization algorithms, which primarily use uncertainty as the basis for this trade-off, our method treats the number of visits to a particular state as a measure of uncertainty. The more frequently a state is visited, the lower its associated uncertainty. This approach is common in Monte Carlo tree search-based methods. We have made a key modification that deviates from traditional settings, enhancing our methodology's effectiveness.
3. **Difference from Reinforcement Learning:** As Reviewer #3 correctly pointed out, our method shares some similarities with goal maximization reinforcement learning algorithms, particularly actor-critic algorithms, where the DNN can be viewed as a critic. However, our approach lacks a dedicated policy network, which distinguishes it from typical reinforcement learning problem settings. Moreover, traditional reinforcement learning methods are often data-intensive, a characteristic often not feasible to be tackled within our problem framework. We also provide evidence to support this distinction.

Below, we present a table that illustrates the working principle of DUCB and its major difference from traditional UCB.

Table R4. The key difference between DUCB and UCB.

From classic upper confidence bound (UCB) to Data-driven upper confidence bound (DUCB)

	Node value	Weight	Value update (Backpropagation)	Exploration
DUCB	 Machine learning Trajectory-independent 	Adaptive	 Short range # of visit only 	Finite value
UCB	 Simulation Trajectory-dependent 	Constant	 Long range # of visit & value 	Infinity everywhere

References:

1. Koide, A. & Koide, S. Monobodies: antibody mimics based on the scaffold of the fibronectin type III domain. *Protein Engineering Protocols* 95–109 (2007).
2. Gebauer, M. & Skerra, A. Engineered protein scaffolds as next-generation therapeutics. *Annu Rev Pharmacol Toxicol* 60, 391–415 (2020).
3. Jumper, J. *et al.* Highly accurate protein structure prediction with AlphaFold. *Nature* 596, 583–589 (2021).

Modifications:

1. We have added the following text passages to the main text portion of the manuscript:

1.1. Summary of our key innovations (page 8 lines 138 to 152):

The Neural-Surrogate-Guided Tree Exploration (NTE) aims at optimizing exploration-exploitation trade-offs through a number of visits and an ML model to deal with non-cumulative reward optimization problems. It resembles the setting of RL, but without the need to train an actor policy network. In the following sections, we explain the working principles of NTE and the rationale behind the newly introduced mechanisms.

We summarize our key innovations as follows:

- A data-driven formula that leverages the number of visits and ML from a small dataset to effectively manage the exploration-exploitation trade-off. This markedly differs from the UCB formula utilized by MCTS, which relies on the average node value and the number of visits derived from a large number of simulations.
- Local backpropagation that ensures a balanced exploration-exploitation trade-off for the non-cumulative reward problems.

- Adaptive exploration mechanism that favors exploration over exploitation under certain circumstances.
- A modified epsilon-greedy sampling technique that samples best-scored candidates and most-visited candidates at the same time.

1.2. Comparison to the Bayesian optimization (page 18 lines 300 to 304):

As indicated in Figure 3a, the BO-based algorithm converges faster than DANTE at low dimensions (<10) and with small initial datasets (<20), while DANTE shows a better performance with higher dimensions and bigger initial datasets. Additionally, Figure 3c shows that a smaller sampling batch size leads to a faster convergence rate. More benchmark results are presented in Supplementary Figures 49-52.

1.3. Difference to the reinforcement learning:

(Page 21 lines 324 to 332) While we consider DANTE and reinforcement learning (e.g., policy proximal optimization (PPO)) to pertain to distinct categories of methodologies in terms of 1) Quantity of data needed; 2) Data accessibility; and 3) Nature of reward (Figure 4a), they can still be compared under specific conditions in the lunar landing task, such as a fixed initial position and random seeds. Under these conditions, DANTE demonstrates comparable, or even better, performance compared to PPO, particularly in the initial stages where PPO essentially performs at a random level, indicating its need for a large amount of data (Figure 4d). However, a notable advantage of PPO is its adaptability, allowing it to be trained for varying environments, such as different initial positions and speeds.

(Page 32 lines 536 to 549) Reinforcement learning (RL) is another commonly used method for identifying optimal solutions. Differences in AL and RL lie in three main aspects: 1) data accessibility, 2) the quantity of data needed, and 3) the nature of rewards (noncumulative vs. cumulative).

- **Data Accessibility:** In typical RL settings, a policy network interacts with the environment, requiring easy access to reward functions. In contrast, active learning, particularly in scientific contexts, often deals with limited access to reward functions. For instance, in materials science, it might take months to obtain just a few labeled data points.
- **Quantity of Data Needed:** RL training commonly demands large amounts of labeled data or observations to develop an effective policy network. AL, however, operates in a low-data regime, usually with fewer than 1,000 data points, and only requires a value-estimation network.
- **Nature of Rewards:** RL algorithms are primarily used for trajectory planning and optimal control problems, involving sequential decisions and cumulative rewards. Conversely, AL typically focuses on maximizing the current reward functions.

2. We have added the following figures to the main text:

Figure 3a and c (Main text). (a) Examples of low-dimensional, small-initial-data benchmark studies are added. (c) Effect of different sampling batch size.

Figure 4a and d (Main text). (a) Comparison of Reinforcement learning and Deep Active Learning. (d) benchmark comparison of DANTE and PPO under specific initial seed.

Figure 5a and b (Main text). TurBO5 (Bayesian-based method) results added to the real-world tasks (structure design).

Figure 5d and e (Main text). TurBO5 (Bayesian-based method) results added to the real-world tasks (complex alloy design).

3. We have added the following sections to the supplementary materials:

Benchmark performance at low dimension and low initial data (page 7 lines 118 to 125)

Real-world task 4: Designing Monoclonal antibody against SARS-COVID-2 (pages 17-18 lines 419 to 448)

‘ In BO terms, the presented optimization algorithm could be viewed as optimizing a fully greedy acquisition function. ’

Response:

Sorry for not making this more explicit. **We discussed the sampling technique in the main text 216-226, and we provide a short summary below:**

As previously mentioned, DANTE employs machine learning predictions to optimize the number of visits to facilitate the exploration-exploitation trade-off. In contrast, Bayesian black-box optimization relies on uncertainty as the primary factor for this trade-off. Our method uses the number of visits to particular states as a measure of uncertainty; the more frequently a state is visited, the lower the uncertainty associated with it. Consequently, in addition to sampling the best-scored candidates, we also sample a portion of random and highly visited candidates. Our ablation study indicates that this approach reduces the number of data points required to achieve global convergence by 30% (see Figure R3b).

‘Especially when targeting applications in the natural sciences, the paper needs a clearer discussion of when or why this approach is advantageous. The general discussion remains vague: is it particularly useful in the low-data regime, in cases of high nonlinearity, or primarily for high-dimensional problems? Why not use a (BO-like) gradient-based optimization of the DNN surrogate?’

Response:

We thank the reviewer for the valuable suggestions. The general discussion remains vague due to the following reasons:

The complexity of an optimization problem can be influenced by its dimensionality and nonlinearity. A rule of thumb of the complexity can be expressed as:

Complexity \propto Dimensionality \times nonlinearity

Therefore, a problem being low-dimensional does not necessarily imply it is easy, particularly in real-world scenarios where the design space may have numerous constraints, exhibit high nonlinearity, and feature noisy labels. For example, in synthetic tasks, a low-dimensional problem (e.g., the Schwefel function with 20 dimensions) that has relatively high nonlinearity can be as challenging as a high-dimensional problem with lower nonlinearity (e.g., the Ackley problem with 2000 dimensions), as both may require a similar number of data points to reach the global optimum.

On the other hand, the definition of a "low-data regime" can vary depending on the context, as different fields have different levels of access to data sources. For our purposes, we have taken the

liberty of defining the low-data regime as having fewer than 200 data points, the medium data regime as between 200 and 1000 data points, and the high data regime as having more than 1000 data points.

On the other hand, **nonlinearity is a challenging term to define precisely**. It generally refers to a function that does not produce a straight line when graphed and lacks a constant slope. We observe that synthetic functions exhibit varying degrees of nonlinearity (Figure R11), and we believe that real-world functions may not be as complex as these synthetic examples. Investigating the mathematical properties of nonlinearity is beyond the scope of this paper.

Figure R11. The benchmark test functions exhibit various degrees of high nonlinearity and are known to be difficult to optimize. However, nonlinearity itself remains hard to define.

Moreover, the issue becomes more pronounced as the dimensionality increases, with the computational costs scaling exponentially. Therefore, due to current hardware limitations, performing active learning on a real-world 1000-dimensional problem is extremely challenging because of the following relationship:

$$\text{Computational cost} \propto \exp(\text{Dimensionality})$$

Overall, we believe that DANTE is a rather versatile and general method capable of addressing a wide range of problems, from low to high dimensions (>10), across various degrees of nonlinearity and from simple to complex systems within the applicable range (initial data points >20). Our empirical observations indicate that DANTE provides significantly improved performance compared to other baseline methods.

To further support our conclusions, we present additional benchmark results for gradient-based Bayesian optimization (BO + CNN) in Figure R13 and R14, which did not demonstrate any advantages. Notably, we observed that BO is exceptionally slow—approximately 50 times slower

in the 100-dimensional benchmark task—compared to our method and other stochastic search methods. This phenomenon (‘curse of dimensionality’) has also been extensively reported in the literature ¹.

The potential combination of testing parameters can go infinitely long and it is not possible to exhaust the list. We provide additional results with **initial 10, 20, 200 data points and difference sampling batches 5, 10, 20**. For test function, **Ackley- 10, 20d; Rosenbrock-10, 20d; Schwefel 10, 20d; Griewank 10, 20** (Figure R13, R14 and R15).

We provide a summary of our result at low dimension and small initial dataset below:

- At a very small initial dataset (<20) and very low dimension (<10), BO-based methods demonstrate a faster convergence rate, but in most cases, DANTE still achieves a better optima (Figure R13, R14 and R15) if sufficient data points are sampled.
- We observe that DANTE converges faster (need fewer data points) with smaller sampling batch size (Figure R12).

Figure R12. BO converges faster than DANTE at low dimension (~10) and small number of initial dataset (~20). While DANTE performs significantly better when the dataset is bigger or dimensional is higher.

Figure R13. Benchmark performance at initial dataset =10. Test functions are Ackley-10d, 20d; Rastrigin-10d, 20d; Rosenbrock-10, 20d; Griewank-10d, 20d; Schwefel-10d, 20d. Selected algorithms: Vanilla BO, TurBO5 (State-of-the-art BO-variant), BO-CNN and DANTE. It can be observed that the BO-based method can often have a faster convergence rate at the beginning. But eventually, as data accumulates, DANTE is able to find a better optima. Notably, as dimensionality increases, DANTE demonstrates clearer advantages.

Figure R14. Benchmark performance at initial dataset =20. Test functions are Ackley-10d, 20d; Rastrigin-10d, 20d; Rosenbrock-10, 20d; Griewank-10d, 20d; Schwefel-10d, 20d. Selected algorithms: Vanilla BO, TurBO5 (State-of-the-art BO-variant), BO-CNN and DANTE. It can be observed that the BO-based method can often have a faster convergence rate at the beginning. But eventually, as data accumulates, DANTE is able to find a better optima. As the number of dimensions considered increases, DANTE demonstrates its advantages more clearly.

Figure R15. Benchmark performance at initial dataset =200. Test functions are Ackley-10d, 20d; Rastrigin-10d, 20d; Rosenbrock-10, 20d; Griewank-10d, 20d; Schwefel-10d, 20d. Selected algorithms: Vanilla BO, TurBO5 (State-of-the-art BO-variant), BO-CNN and DANTE. DANTE demonstrates a clear advantage over the other algorithms tested.

Modifications:

We have now added the following section to the supplementary materials:

Benchmark performance at low dimension and low initial data (page 7 lines 118 to 125)

‘In this context, I find especially Fig. 3a and the related discussion somewhat misleading. If sample efficiency is a main concern (in relatively low-dimensional spaces), is there evidence that deep active learning is superior? I would argue that for low-dimensional, low-data problems, classical BO (e.g. using Gaussian Process surrogates) can be more sample efficient, because the surrogate model can provide good predictivity with as few as 10–20 examples. In this work, the authors always initialize their model with 200 observations, which is rather high for many real-world experimental settings. Although I appreciate the idea of the plot, the figure and the text must be much clearer about in which regimes the DAL actually constitutes an advantage.’

Response:

We provide additional evidence on the performance of low dimensional low data problems (defined as dimension < 10 and data point < 20) in the previous response.

‘A second main concern is that the workflow does not address the exploration–exploitation tradeoff when optimizing the unknown objective function(similar to a “fully greedy acquisition function optimization”, see above). In classical active learning, this is often proposed to be crucial for enabling sample-efficient optimization. Is this a problem for experimental use cases? In the manuscript, this possible limitation is hidden behind a very detailed discussion of the algorithm that is used for optimizing the “acquisition function” (i.e. the DNN predictions) over the full search space. Would a combination with cheap uncertainty estimates for DNN models (e.g. Laplace approximation or variational inference) be feasible to circumvent this?’

Response:

We thank the reviewer for their valuable suggestions. We would like to emphasize that **the tree search method represents a distinct approach that primarily utilizes search history for the exploration-exploitation trade-off**. Specifically, the number of visits to a particular state serves as a proxy for the uncertainty associated with that state;

For instance, the more frequently a state is visited, the lower its uncertainty. DANTE leverages this information to effectively balance exploration and exploitation rates. Additionally, the method incorporates adaptive exploration strategies to prioritize exploration over exploitation under specific circumstances. All these mechanisms contribute to the exploration-exploitation trade-off and are discussed in the result sections.

A combination of uncertainty derived from an ensemble model has been proposed alongside the original UCB formula; however, it has not been widely adopted in subsequent research ². This is likely due to the fact that the visit history alone provides sufficient exploration. Nevertheless, the reviewer raises an interesting question that warrants further exploration, although we believe it falls outside the scope of the current study.

To increase readability of the paper, we significantly modified our introduction by providing now a more thorough discussion on various genres of learning-based optimization methods and incorporating a brief summary at the beginning of the result section and dissect the entire section into four different subsections: Conditional selection; Local backpropagation; Data-driven upper confidence bound and Adaptive exploration.

Modifications:

We have added the following sections to the supplementary materials:

Benchmark performance at low dimension and low initial data (page 7 lines 118 to 125)

‘DNN architecture: How is it chosen? Is generalizable to different problem types, or are problem-specific models needed?’

Response:

The selection of DNN is rather straightforward, and we use the DNN to fit the initial dataset and see if it performs well on the testing dataset (Pearson ratio > 0.75). If it does not do so, we perform additional hyperparameter tuning on the architecture until it does. Overall, we maintain a very similar backbone throughout the whole project. For more details, please see the supplemental materials “Supplementary Notes”.

Specifically, for the evaluations on synthetic function and neural architecture search (NAS) task, we used a 1D-CNN which comprises ~ 5 convolutional layers with filter sizes of 128, 64, 32, 16, and 8 respectively, each using a kernel size of 3. With ~ 2 max-pooling layers with a pooling size of 2, ~ 2 dropout layers with a dropout rate of 0.2, followed by a flatten layer, 2 fully connected layers with 128 and 64 units respectively, and an output layer. The loss function is MSE, learning rate for the Adam Optimizer is set at 0.001, and the activation function utilized is the Exponential Linear Unit (ELU). The 1D-CNN model is trained for 500 epochs with an early stopping patience of 30, and a batch size of 64.

As to the soft magnetic alloy real world task test, compositionally complex alloy design, and other tasks with 1-dimensional input features, we also used a 1D-CNN but with different model architecture and hyperparameters.

For the example dealing with architected materials design, we used a 3D-CNN model that comprises 3 convolutional layers with filter sizes of 8, 4, and 2 respectively, each using a kernel size of 3. Each convolutional layer is followed by a max-pooling layer with a pooling size of 2. Before the output layer, there is a flatten layer and 3 fully connected layers with 128, 64, and 32 units respectively. The loss function is MSE, the learning rate for the Adam Optimizer is set at 0.001, and the activation function is ELU. The 3D-CNN model is trained for 5000 epochs with an early stopping patience of 100, and a batch size of 32.

‘ - Metrics: The authors primary evaluate the number of experiments required to find the global optimum. Is this truly the most relevant metric? For many scientific applications, identifying the single best optimum is less important than discovering several top solutions.’

Response:

We argue that the number of experiments (or data) is the most relevant metric from an experimentalist's perspective. As the majority of our project members have extensive experimental/computational background, we recognize that the bottleneck in the process of experimental/computational discovery of new materials is primarily the cost of experimentation. Therefore, we always look for a method capable of finding optimal solutions with minimal required data and thus minimal additional workload from experiments, which - particularly in the area of complex and real-world materials problems - is often expensive and lengthy. More specific, current experimental costs for developing, synthesizing, processing and optimizing a single real material can reach hundreds of thousands of dollars and require years of labor, encompassing the processes of purchasing, preparation, and characterization of the candidate materials. For high-dimensional computational tasks, one single run might cost millions of CPU hours on the cluster. Therefore, in most cases, we prioritize the method that can find the best solution using minimal data points and selecting candidates that exhibit the best performance. If the reviewer is interested in identifying multiple top solutions, we can easily adjust the setting from a top-1 search to a top-k search.

‘ - Dimensionality of synthetic and experimental benchmarks: The authors demonstrate success on high-dimensional benchmark surfaces, but the “real-life” scientific examples have significantly lower dimensionality. How would BO-type methods perform in these cases?’

Response:

The results of the Bayesian optimization (TurBO5) method have now been included (Figure R16), which is an improved variant of traditional Bayesian optimization. However, we find that both these methods do not rank among the top-performing approaches in the here addressed task portfolio of high dimensionality and nonlinearity. As previously explained, the complexity of an optimization problem is highly influenced by both dimensionality and nonlinearity, and even low-dimensional real-world problems can be quite difficult to optimize due to the high nonlinearity, external constraints, and noisy labels.

Figure R16. Results from TurBO5, due to the high-nonlinearity, various constraints and noise, external constraints, TurBO5 (State-of-the-art BO variant) 's solution is worse than Dante's.

Modifications:

We have added the following figures to the main text:

Figure 5 a, b, d and e. (details see previous reply)

'- Peptide example: The featurization is based on one-hot encoding, so the distances do not reflect any physical meaning. Some of the expansion policies become unclear in this context.'

Response:

Thank you for the comment. We appreciate the opportunity to clarify this point. In our cyclic peptide design example, one-hot encoding is used solely for parameterizing the peptide sequences. The expansion process is not based on any distance metric between sequences. Instead, our approach employs two independent strategies: random mutations applied to selected random sites, and the generation of entirely new, random sequences. In our modeling rounds, these two methods were used in a relative ratio of 2:1. We have updated the manuscript to make this distinction clearer.

'- Batch-wise evaluation: In their experiments, the authors show few iterations with batch sizes of 20 experiments per iteration. What is the computational cost of generating each batch of recommendations? Would a smaller batch size yield improved sample efficiency?'

Response:

For computational cost, the biggest problem is that computational cost scales with the dimension. A rule of thumb is that:

Computational cost $\propto \exp(\text{Dimensionality})$

This relation holds true for most real-world applications. To better illustrate our standpoint, we provide now a more general discussion about the FEM simulation related to real-world applications:

For the case study of scaffold design (Figure 5 a,b,c in the main text), we performed the compression simulation on a 32-core, 64-thread CPU (Intel Xeon Gold 6226R Processor) using the commercial ABAQUS/Explicit software, one of the global market leaders in this field. Specifically, the computational resources consumed during the compression process are listed in Table R5. With a constant compression strain and mesh size, the computational resources required for a 5×5×5 scaffold increased exponentially. This is due to the cubic relationship between the number of mesh elements and the geometric dimensions. Additionally, the increased compression displacement further extended the computation time. When designing a small animal implant scaffold as shown in Figure R17, the simulation time exceeded one day. In actual clinical applications, after a patient's surgical plan is finalized, the design and manufacturing of the implant typically need to be completed within a month. Therefore, the computational cost becomes the bottleneck which requires improvement on the FEM software itself.

Finally, we tested the effects of different sampling sizes, and the results are presented in Figure R18. Interestingly, smaller sampling batch sizes lead to a faster convergence rate. This finding has now also been included into the main text of the revised paper.

Figure R17. A real-world case study involving bone implant design illustrates that costs increase exponentially with the dimensionality of the design parameters.

Table R5. The comparison of computational costs across varying dimensions of the bone implant problem reveals that both the time required for computation and the number of elements involved escalate dramatically as the dimensionality increases.

Scaffold	Dimension	Compression strain (%)	Number of elements	Time budget (min)
3*3*3	27	6	112085	10.8
5*5*5	125	6	473374	67.2
Rabbit bone	210	6	2019673	1545.6

Figure R18. DANTE with different sampling batch sizes. The smaller sampling batch size could lead to faster convergence rate.

Modifications:

1. We have now added the following figures to the main text:

Figure 3a and c in main text.

2. We have now added the following section to the supplementary materials:

Effects of different sampling batch size (page 7 lines 126 to 129)

Reviewer #2:

‘1.The authors claim that, different from the MCTS method used in examples like AlphaGo, they propose a method incorporates novel mechanisms such as UCB. This is not fair, it is known that

UCB is a popular utility function and has been used in many optimization problem. The author should clarify what is the novelty in method compared to previous ones.'

Response:

We cordially thank the reviewer for the valuable suggestions. The mechanism of DUCB is very different from the traditional UCB (as also pointed out by Reviewer #3 and Reviewer #4).

It is essentially a data-driven, adaptive formula that leverages **the number of visits and specific type of DNN model from a small dataset** to effectively manage the exploration-exploitation trade-off. This approach markedly differs from the UCB formula utilized by MCTS, which relies on the **average node value and the number of visits derived from a large number of simulations**. The key difference is illustrated in Table R6 below:

Table R6. The key difference between DUCB and UCB

From classic upper confidence bound (UCB) to Data-driven upper confidence bound (DUCB)

	Node value	Weight	Value update (Backpropagation)	Exploration
DUCB	 Machine learning Trajectory-independent 	Adaptive	 Short range # of visit only 	Finite value
UCB	 Simulation Trajectory-dependent 	Constant	 Long range # of visit & value 	Infinity everywhere

Overall, DANTE is a hybrid methodology that synthesizes features from various optimization genres, enriched by key modifications that result in a unique and innovative variant.

We summarize our key innovations as follows:

- **A data-driven formula that leverages the number of visits and ML from a small dataset to effectively manage the exploration-exploitation trade-off.** This markedly differs from the UCB formula utilized by MCTS, which relies on the average node value and the number of visits derived from a large number of simulations.
- **Local backpropagation** that ensures a balanced exploration-exploitation trade-off for the non-cumulative reward problems and accelerates the search for maximal non-cumulative reward.
- **Adaptive exploration mechanism** that favors exploration over exploitation under certain circumstances.
- **A modified epsilon-greedy sampling technique** that samples best-scored candidates and most-visited candidates in one batch.

To increase the readability, we significantly modified our introduction by introducing a thorough discussion on various genres of learning-based optimization methods and incorporating a brief summary at the beginning of the result section. We move the ‘framework of active learning’ to the Method section and dissect the entire section into four different subsections: Conditional selection; Local backpropagation; Data-driven upper confidence bound and Adaptive exploration. We also provide a comprehensive comparison between Bayesian methods and Reinforcement learning methods.

Modifications:

We have now added the following text items to the main text:

Summary of our key innovations (page 8 lines 138 to 152):

The Neural-Surrogate-Guided Tree Exploration (NTE) aims at optimizing exploration-exploitation trade-offs through a number of visits and an ML model to deal with non-cumulative reward optimization problems. It resembles the setting of RL, but without the need to train an actor policy network. In the following sections, we explain the working principles of NTE and the rationale behind the newly introduced mechanisms.

We summarize our key innovations as follows:

- A data-driven formula that leverages the number of visits and ML from a small dataset to effectively manage the exploration-exploitation trade-off. This markedly differs from the UCB formula utilized by MCTS, which relies on the average node value and the number of visits derived from a large number of simulations.
- Local backpropagation that ensures a balanced exploration-exploitation trade-off for the non-cumulative reward problems.
- Adaptive exploration mechanism that favors exploration over exploitation under certain circumstances.
- A modified epsilon-greedy sampling technique that samples best-scored candidates and most-visited candidates at the same time.

‘2. One of the key issues the manuscript plans to address is the high-dimensional space, for example, the one with 2000 dimensions. They demonstrate their method in several scenarios. However, it is questionable if this is the real case in practical applications. At least in materials science, the dimension is often not as high as 2000. More importantly, for high dimensions feature engineering is usually first used to eliminate the features with poor information, which can help simplify the surrogate model, increase the interpretability, and save computational resource. So it is difficult to imagine why the authors want to apply the method to such a high dimensional space.’

Response:

We thank the reviewers for the most valuable suggestions and comments. Indeed, we argue that high-dimensional optimization is very interesting and potentially very useful from two specific perspectives:

- 1) It is interesting to see that DANTE is able to converge at such a high numbers of dimensions. A deeper theoretical explanation to this would indeed be very interesting.
- 2) Many real-world tasks can be formulated as high dimensional problems. What really becomes problematic is that as the dimensionality increases, the computational cost scales up exponentially. A rule of thumb is that ³:

Computational cost $\propto \exp(\text{Dimensionality})$

Therefore, limited by current hardware and the awareness of the pressing need to conduct AI tasks more economically and also in a more sustainable fashion, one could hardly perform an active learning with a real world 1000-dimensional problem. Nonetheless, we provide two real-worlds cases where high-dimensional optimization could be very handy if sufficient resources are available, 1) high-dimensional alloy design 2) high-dimensional metamaterials design.

1. High-dimensional alloy design

Again, we agree with the reviewer that in the previous literature when handling high-dimensional data, the feature engineering, like dimension-reduction strategy PCA, is more often adopted. However, we would like to stress that the employment of feature engineering would lead to the loss of structural recovery capability. In this regard, to ensure simultaneously good representation of the material and recoverability, a high-dimension input is usually necessary. Here, we use the optimization of ice polymorphs as an example (Figure R19). Despite its simple molecular composition, ice exhibits a large number of polymorphs ⁴, covering at least 20 known crystalline phases and various amorphous material forms. Discovering new ice polymorphs is crucial in quite diverse fields, e.g., astrobiology, climate & glacier science, exoplanet research, food industry, and energy storage, to name but a few interesting application fields. Unfortunately, the discovery of new ice polymorphs is a rather challenging task due to the large configurational space. Nevertheless, we believe this is beyond the current scope of this manuscript as computing the property of ice structure falls out of our expertise. The domain expert can utilize the power of the DANTE algorithm to optimize these high-dimensional structural descriptors.

Figure R19. Ice structure optimization may require more than 500 structural descriptors, as utilizing such high-dimensional descriptors facilitates a more comprehensive exploration of the ice structure space, enabling better recovery of the ice structure.

First, it has been realized that one can train a machine learning interatomic potential for ice⁵, which can be used to replace the role of, e.g., DFT, to give the energy of a large-scale or complex ice structure. To achieve the goal of ice structure optimization, the ice structures will be represented through high-dimensional structural descriptors, such as, the radial distribution function (RDF) among species for capturing atomic distances and coordination shells, the bond order parameters (BOPs) for encoding tetrahedrality and local ordering, and the angular distribution function (ADF) for describing bond angles. The reasons why these descriptors have been selected here include: (1) these high-dimensional descriptors provide a more comprehensive representation of atomic interactions; and more importantly (2) it is possible to use such high-dimensional descriptors to recover the ice structure, thus allowing a more complete exploration of ice structure space. In specific, one can utilize the inverse structure prediction methods, for instance, the reverse Monte Carlo (RMC) fitting to reconstruct atomic configurations with these descriptors.

In terms of the dimensionality, the RDF ($g(r)$) describes the probability of finding a particle at a distance r (with cutoff around 5~10 Å) from a reference atom. The representation of this function requires the discretization of r , constituting 100 – 200 dimensions for a better resolution and for the account of long-range interactions. The BOPs quantify the local structural order around an atom, which is defined by a spherical harmonics series expansion to an order of l_{max} . The dimension of the BOPs depends obviously on the choice of the truncation value l_{max} . Common choices of l_{max} are 6 - 8, and its dimension can reach up to 100. The ADF measures the angle distributions between the bonded atoms, which is particularly relevant for hydrogen bonding networks in ice. Similar to RDF, the distribution function is discretized to 150 – 300 bins. **Overall,**

the dimension of the relevant descriptors can easily exceed 500. Particularly, in some cases, it might be crucial to go to higher dimensions, in order to capture the long-range interactions in ice structures and to ensure better structural recovery.

2. High-dimensional metamaterials design

We would argue that the high-dimensional design (optimization) problem in material science is an important question to address (Figure R20 as example).

Figure R20. Bone implant design - a real world optimization problem, the task is to design the size and shape of hundreds of individual cells to achieve better mechanical properties. It is a typical high-dimensional optimization problem, even though the cell geometry has already been simplified. from reference ⁶.

Table R7. Comparison of computational cost given different dimensions of the bone implant problem. Both time cost and number of elements explode as the dimension increases.

Scaffold	Dimension	Compression strain (%)	Number of elements	Time budget (min)
3*3*3	27	6	112085	10.8
5*5*5	125	6	473374	67.2
Rabbit bone	210	6	2019673	1545.6

In structural design shown in Figure R20 and Table R7, each unit represents a dimension. For example, a $3 \times 3 \times 3$ scaffold consists of 27 dimensions. However, as shown in Figure R20, in the practical application of small animal bone implants, the structure size is 10×20 mm, resulting in approximately 210 dimensions. If applied to human clinical use in the future, the size will increase exponentially. For instance, an implant with a diameter of 20 mm and a length of 40 mm would require around 2000 dimensions. Moreover, every unit in this structure is essential; otherwise, the structure would be incomplete. Therefore, optimization in a high-dimensional space is inevitable in this case.

Besides, feature engineering has its own limitation, because the feature importance is often a result of information richness pertaining to a specific property. For example, in high entropy alloy design of the resulting INVAR properties (alloys with very low thermal expansion coefficient), when running a feature importance analysis on the available dataset, the importance of copper often come as the last⁷. However, this is not true because from our domain knowledge and from basic physics knowledge about the relationship between ferromagnetism and phonon-driven thermal expansion, we know that it is indeed a quite essential element and its lack of information richness is mainly due simply to a profound lack of exploration by human experts of the role of this element in these materials. But if we trust feature importance alone, this will result in elimination of copper as a designable parameter, which is undesirable because we are quite interested in material tweaking effects potentially offered by this element.

Real-world task 4: Designing Monoclonal antibody against SARS-COVID-2

Finally, while high-dimensional deep active learning is computationally expensive, recent advancements in protein structure prediction software, such as AlphaFold 3, have significantly accelerated the interpolation process. This progress enables us to introduce a pertinent high-dimensional example of active learning in the context of protein design, which involves optimizing hundreds of known amino acids. As achieving this optimization using AlphaFold 2, as utilized in our current project, presents considerable challenges. We present an example involving hundreds of dimensions, with AlphaFold 3 serving as the validation source. In our analysis, we also tested several baseline methods, including TurBO (BO-variant), MCMC, and DANTE.

Figure R21. Pipeline of designing monobody binder to SARS-COVID-2 spike protein. Search strategies are DANTE, MCMC and TurBO5.

Monoclonal antibodies (mAbs) are among the most common therapeutics and are frequently used for the treatment of cancer, autoimmune diseases, infectious diseases, and more. Recently, monobodies (Mobs) have gained attention as an alternative to mAbs due to their strong developability^{8,9}. Mobs are proteins roughly 100 amino acid residues in length, featuring an immunoglobulin (Ig)-like fold composed of 7 beta strands, derived from human Fibronectin type III (FN3). These proteins combine the advantageous features of both mAbs and miniproteins, which are generated through de novo design approaches, while also overcoming some of their disadvantages.

We build the Monobody binder design pipeline using DANTE, MCMC and TuRBO5 targeting the SARS-COVID-2 spike protein. Similar to the real-world tasks proposed in the real-world task 3 in the main text, the pipeline uses AlphaFold 3, Rosetta and ProteinMPNN to generate pools of monobody binders with high ipTM score.

ipTM is a metric used to assess the accuracy of predicted relative positions in protein-protein complexes¹⁰. Values greater than 0.8 indicate confident high-quality predictions, suggesting that the model has likely captured the correct orientation and interaction of the subunits. Conversely, values below 0.6 suggest a likely failed prediction, indicating potential inaccuracies in the model. ipTM values falling between 0.6 and 0.8 are considered a grey zone, where predictions could either be correct or erroneous, requiring further validation or additional evidence to ascertain their reliability.

The ipTM distributions obtained from different methods—DANTE, TurBO5, and MCMC—reveal that DANTE significantly outperforms TurBO5 and generates a greater number of high-quality designs compared to MCMC. This indicates that DANTE is more effective at predicting the accurate relative positions of subunits in protein-protein complexes, leading to more reliable and successful design outcomes.

Figure R22. The ipTM distributions obtained from different methods—DANTE, TurBO5, and MCMC—reveal that DANTE significantly outperforms TurBO5 and generates a greater number of high-quality designs compared to MCMC.

Table R8. Percentile of high-quality solutions designed by three methods.

Methods	Percentile of high-quality solutions (ipTM>0.8)
DANTE	27%
MCMC	7%
TurBO5	0%

‘3. One page 3, “Traditional machine learning models, such as Bayesian methods, heavily rely on assumptions about prior distributions or feature engineering”. This statement is too strong, there are many machine learning models, not all the models have assumptions on data distribution etc.’

Response:

We thank reviewers for the valuable suggestions, we made the corresponding changes in the main text.

Modifications:

We have now added the following text portions to the main text (page 3 lines 56 to 59):

‘Some machine learning (ML) models, such as Bayesian methods, heavily rely on assumptions about prior distributions or feature engineering, while other models, like decision trees, are prone to overfitting and can only process specific types of data, such as tabular formats.’

'4.The organization of the manuscript is too wordy, e.g., the introduction of AL on page 3 and page 7.'

Response:

We thank the reviewers for the valuable suggestions. We fully comply and we reorganized the whole narrative, with the section on page 3 going to the 'Methods' section.

'5.The authors first compare different surrogate models and find DNN is the fastest, then different selectors are compared and the DA outperforms the others. Can this process ensure that DNN+DA is the best, as not all the possible combinations are examined.'

Response:

We acknowledge that the question posed is inherently broad, making it challenging to cover all potential combinations exhaustively. Our investigation, focusing on the synergy between Deep Neural Networks (DNN) and Deep Active Learning (DAL), was conducted with an emphasis on simplicity. Starting with straightforward surrogate models across selected benchmark tests, we observed that DNNs exhibited superior performance compared to alternative models.

We recognize the potential interest in exploring other surrogate models, such as transformers; however, this exploration is beyond the primary focus of our current manuscript, which is centered on the mechanism of tree search. Moreover, our existing results already demonstrate state-of-the-art performance.

Regarding other baseline comparisons involving different surrogate models, we utilized their default settings as established in relevant prior studies^{11,12}. We believe that our testing encompasses all significant surrogate model types frequently referenced in mainstream literature.

Reviewer #3

'1. My main comment is related to the classification of DANTE as an active learning method. I would suggest that the authors revise this decision upon deeper review of the reinforcement learning literature. From the presentation of the algorithm, it seems that the only connection to active learning is that a few promising samples are selected for the training of the surrogate model after each iteration. Active learning typically has the goal of reducing the error or uncertainty in the trained model while labeling the least number of data possible. There are many examples available in the literature, [1] and [2] are just some recently published ones. In the unnumbered formula on line 146, page 8, the authors seem to interpret the active learning selection criterion as simply maximizing the value from the surrogate model, rather than making any attempts to quantify uncertainty, effect of the sample on the model, or other metric to select maximally

informative samples. On the other hand, this understanding aligns very well with the reward-maximization goal of reinforcement learning. Of course, there are many variations in machine learning methods and the differences between one framework and the other may blur. Nonetheless, DANTE draws upon decades of reinforcement learning developments (tree search, UCB, greedy-epsilon, value-estimation networks, AlphaGo) and almost nothing on active learning theory.'

Response:

Many thanks indeed. This is an excellent comment, and we fully agree with the reviewer that traditional active learning typically aims to reduce error or uncertainty in the trained model, while minimizing the number of labeled data points. As the reviewer correctly pointed out, "*unlike typical active learning settings, the selection score for the samples is the same objective function to be optimized.*" We also believe that optimization should be included as a critical metric of active learning, as high predictive accuracy from the surrogate model does not necessarily guarantee effective optimization. This consideration is particularly crucial when targeting the field of scientific discovery, where the goal is to identify candidates with superior performance in an iterative manner.

We believe the key differences between active learning (AL) and reinforcement learning (RL) lie in three main aspects: 1) data accessibility, 2) the quantity of data needed, and 3) the nature of rewards (noncumulative vs. cumulative).

1. **Data Accessibility:** In typical RL settings, a policy network interacts with the environment, requiring easy access to reward functions. In contrast, active learning, particularly in scientific contexts, often deals with limited access to reward functions. For instance, in materials science, it might take months to obtain just a few labeled data points.
2. **Quantity of Data Needed:** RL training commonly demands large amounts of labeled data or observations to develop an effective policy network. AL, however, operates in a low-data regime, usually with fewer than 1,000 data points, and only requires a value-estimation network.
3. **Nature of Rewards:** RL algorithms are primarily used for trajectory planning and optimal control problems, involving sequential decisions and cumulative rewards. Conversely, AL typically focuses on maximizing the current reward functions.

To comprehensively compare AL with RL, we provide the following plots:

Figure R23. Comparison of DAL and RL in terms of quantity of data needed, data accessibility and nature of rewards.

In summary shown in Figure R23, DANTE is a hybrid methodology that combines features from three distinct genres of optimization methods, incorporating key modifications that result in a unique mixed-type approach.

Modifications:

- 1. We have extended our introduction to RL and included various references.**

Difference to the Reinforcement learning:

(Page 21 lines 324 to 332) While we consider DANTE and reinforcement learning (e.g., policy proximal optimization (PPO)) to pertain to distinct categories of methodologies in terms of 1) Quantity of data needed; 2) Data accessibility; and 3) Nature of reward (Figure 4a), they can still be compared under specific conditions in the lunar landing task, such as a fixed initial position and random seeds. Under these conditions, DANTE demonstrates comparable, or even better, performance compared to PPO, particularly in the initial stages where PPO essentially performs at a random level, indicating its need for a large amount of data (Figure 4d). However, a notable advantage of PPO is its adaptability, allowing it to be trained for varying environments, such as different initial positions and speeds.

(Page 32 lines 536 to 549) Reinforcement learning (RL) is another commonly used method for identifying optimal solutions. Differences in AL and RL lie in three main aspects: 1) data accessibility, 2) the quantity of data needed, and 3) the nature of rewards (noncumulative vs. cumulative).

- **Data Accessibility:** In typical RL settings, a policy network interacts with the environment, requiring easy access to reward functions. In contrast, active learning, particularly in scientific contexts, often deals with limited access to reward functions. For instance, in materials science, it might take months to obtain just a few labeled data points.
- **Quantity of Data Needed:** RL training commonly demands large amounts of labeled data or observations to develop an effective policy network. AL, however, operates in a low-data regime, usually with fewer than 1,000 data points, and only requires a value-estimation network.
- **Nature of Rewards:** RL algorithms are primarily used for trajectory planning and optimal control problems, involving sequential decisions and cumulative rewards. Conversely, AL typically focuses on maximizing the current reward functions.

2. We have now added the following figures to the main text:

Figure 4a and d (Main text). (a) Comparison of Reinforcement learning and Deep Active Learning. (d) benchmark comparison of DANTE and PPO under specific initial seed.

'2. The issue of presenting the method as an active learning technique is not merely a semantic one. The lack of discussion of reinforcement learning also means that comparisons against other methods may conveniently omit some techniques. In particular, in line 323, page 19, the authors write "DANTE significantly outperforms other AL methods" but there seems to be no effort to compare against reinforcement learning techniques. For example, why was soft actor-critic [3] not used in comparison for the lunar landing benchmark where it is possible to frame the problem as a cumulative reward objective? Wouldn't an algorithm that performs sequential decision making be more suitable in that case?'

Response:

While we consider DANTE and PPO to belong to distinct categories of methodologies, they can still be compared under specific conditions, such as a fixed initial position and random seeds. Under these conditions, DANTE demonstrates comparable, or even better performance compared to PPO, particularly in the initial stages where PPO essentially performs at random (Figure R24). However, a notable advantage of PPO is its adaptability, allowing it to be trained for varying environments, such as different initial positions and speeds. In contrast, any alteration in the environment necessitates the re-training or re-running of DANTE. For detailed experimental setups, please refer to the main text.

Figure R24. performance comparison (X-axis in normal timestep scale and log timestep scale) of DANTE and PPO under fixed position and random seed,

Modifications:

We have now added the following figures to the main text:

Figure 4a and d in main text.

‘3. In order to revise the manuscript, I think the authors need to provide a deeper discussion of reinforcement learning works so that readers can compare DANTE with other methods that apply similar ideas. It might be helpful to explain the distinguishing features between active and reinforcement learning in the introduction to clarify why the authors place the method in one category or the other. Finally, I think that a comparison to at least one modern reinforcement learning method must be provided for at least one of the benchmarks.’

Response:

Yes, we have provided a thorough discussion in the introduction as well as PPO in the main text Figure 4a and d in the main text. For discussion, we kindly refer to the previous reply. The suggested references have been added to the main text.

Modifications:

Added references:

- [1] Kulichenko, Maksim, et al. "Uncertainty-driven dynamics for active learning of interatomic potentials." *Nature Computational Science* 3.3 (2023): 230-239.
- [2] Yang, Jason, et al. "Active learning-assisted directed evolution." *Nature Communications* 16.1 (2025): 714.
- [3] Haarnoja, Tuomas, et al. "Soft actor-critic: Off-policy maximum entropy deep reinforcement learning with a stochastic actor." *International conference on machine learning*. Pmlr, 2018.

Reviewer #4 (Remarks to the Author):

'This suggests that DANTE can also handle multi-objective optimization problems. The authors are encouraged to discuss this aspect explicitly in the manuscript and provide additional multi-objective case studies (in the Supplementary). Moreover, are there limitations regarding the number of objectives the method can handle? Can it effectively discover the Pareto front in such cases?'

Response:

Thank you for your positive feedback and kind words. We recognize that most real-world problems are inherently multi-objective. To address this in our project, we convert multi-objective problems into single-objective ones by combining multiple objectives into a single metric. This approach has been applied in two tasks: the design of magnetic alloys and cyclic peptides, with results illustrated in Figure R25. The best Pareto front solution has been identified using DANTE. We are aware that there are many other ways to search for Pareto front, this one seems to be working fine in our cases.

Figure R25. (From main text Figure 4 and 5), two multi-objective alloy design tasks, and the stars represent the Pareto front solutions discovered by DANTE.

Recent advancements in protein structure prediction software, such as AlphaFold 3, have led to substantial increases in computational efficiency. This progress enables us to introduce a pertinent high-dimensional example of active learning in the context of protein design, which involves optimizing hundreds of amino acids. As achieving this optimization using AlphaFold 2, as utilized in our current project, presents considerable challenges. We present an example involving hundreds of dimensions, with AlphaFold 3 serving as the validation source. In our analysis, we also tested several baseline methods, including TurBO (BO-variant), MCMC, and DANTE.

Figure R26. Pipeline of designing monobody binder to SARS-COVID-2 spike protein. Search strategies are DANTE, MCMC and TurBO5.

Monoclonal antibodies (mAbs) are among the most common therapeutics and are frequently used for the treatment of cancer, autoimmune diseases, infectious diseases, and more. Recently, monobodies (Mobs) have gained attention as an alternative to mAbs due to their strong

developability^{8,9}. Mobs are proteins roughly 100 amino acid residues in length, featuring an immunoglobulin (Ig)-like fold composed of 7 beta strands, derived from human Fibronectin type III (FN3). These proteins combine the advantageous features of both mAbs and miniproteins, which are generated through de novo design approaches, while also overcoming some of their disadvantages.

We build the Monobody binder design pipeline using DANTE, MCMC and TuRBO5 targeting the SARS-COVID-2 spike protein. Similar to the real-world tasks proposed in the real-world task 3 in the main text, the pipeline uses AlphaFold 3, Rosetta and ProteinMPNN to generate pools of monobody binders with high ipTM score.

ipTM is a metric used to assess the accuracy of predicted relative positions in protein-protein complexes¹⁰. Values greater than 0.8 indicate confident high-quality predictions, suggesting that the model has likely captured the correct orientation and interaction of the subunits. Conversely, values below 0.6 suggest a likely failed prediction, indicating potential inaccuracies in the model. ipTM values falling between 0.6 and 0.8 are considered a grey zone, where predictions could either be correct or erroneous, requiring further validation or additional evidence to ascertain their reliability.

The ipTM distributions obtained from different methods—DANTE, TurBO5, and MCMC—reveal that DANTE significantly outperforms TurBO5 and generates a greater number of high-quality designs compared to MCMC. This indicates that DANTE is more effective at predicting the accurate relative positions of subunits in protein-protein complexes, leading to more reliable and successful design outcomes.

Figure R27. The ipTM distributions obtained from different methods—DANTE, TurBO5, and MCMC—reveal that DANTE significantly outperforms TurBO5 and generates a greater number of high-quality designs compared to MCMC.

Table R9. Percentile of high-quality solutions designed by three methods.

Methods	Percentile of high-quality solutions (ipTM>0.8)
---------	---

DANTE	27%
MCMC	7%
TurBO5	0%

‘While the manuscript is well-written, the section “Neural-Surrogate-Guided Tree Exploration” is convoluted and lacks clarity. Improving the flow of this section, as well as the overall readability of the manuscript, would enhance comprehension.

Additionally, in Eq. (1), the variables N and n are not introduced or explained. Clarifying their definitions would improve the mathematical clarity of the paper.’

Response:

Thanks for the suggestions, to increase the readability, we significantly modified our introduction by introducing a thorough discussion on various genres of learning-based optimization methods and incorporating a brief summary at the beginning of the result section. We move the ‘framework of active learning’ to the Method section and dissect the entire section into four different subsections: Conditional selection; Local backpropagation; Data-driven upper confidence bound and Adaptive exploration. We also provide a comprehensive comparison between Bayesian methods and Reinforcement learning methods.

Modifications:

We have changed the main text as follows (page 10 lines 187-188):

N is the number of current root nodes and n is the number of visits of the current node.

References:

1. Snoek, J., Larochelle, H. & Adams, R. P. Practical bayesian optimization of machine learning algorithms. *Adv Neural Inf Process Syst* **25**, (2012).
2. Auer, P., Cesa-Bianchi, N. & Fischer, P. Finite-time analysis of the multiarmed bandit problem. *Mach Learn* **47**, 235–256 (2002).
3. Marin, F., Souza, A. F. de, Pabst, R. G. & Ahrens, C. H. Influences of the mesh in the CAE simulation for plastic injection molding. *Polimeros* **29**, e2019043 (2019).
4. Salzmann, C. G. Advances in the experimental exploration of water’s phase diagram. *J Chem Phys* **150**, (2019).

5. Zivkovic, A., Terranova, U. & de Leeuw, N. H. Water Is Cool: Advanced Phonon Dynamics in Ice Ih and Ice XI via Machine Learning Potentials and Quantum Nuclear Vibrations. *J Chem Theory Comput* (2025).
6. Peng, B. *et al.* Machine learning-enabled constrained multi-objective design of architected materials. *Nat Commun* **14**, 6630 (2023).
7. Rao, Z. *et al.* Machine learning-enabled high-entropy alloy discovery. *Science* (1979) **378**, 78–85 (2022).
8. Koide, A. & Koide, S. Monobodies: antibody mimics based on the scaffold of the fibronectin type III domain. *Protein Engineering Protocols* 95–109 (2007).
9. Gebauer, M. & Skerra, A. Engineered protein scaffolds as next-generation therapeutics. *Annu Rev Pharmacol Toxicol* **60**, 391–415 (2020).
10. Jumper, J. *et al.* Highly accurate protein structure prediction with AlphaFold. *Nature* **596**, 583–589 (2021).
11. Wang, L., Fonseca, R. & Tian, Y. Learning search space partition for black-box optimization using monte carlo tree search. *Adv Neural Inf Process Syst* **33**, 19511–19522 (2020).
12. Eriksson, D., Pearce, M., Gardner, J., Turner, R. D. & Poloczek, M. Scalable global optimization via local Bayesian optimization. *Adv Neural Inf Process Syst* **32**, (2019).

Response to reviewer's comments

Reviewer #1 (Remarks to the Author):

'Additionally, I thank the authors for clarifying my misinterpretation regarding the role of the tree search algorithm. I found the statement that it manages the exploration–exploitation tradeoff in a frequentist manner very convincing, and I feel that the clarity of the manuscript could benefit from a similar explanation, too.'

Response:

We thank the reviewer very much for the strong support! We are delighted that our explanations have helped to clarify the concerns. We fully comply to this suggestion and we have added this part to the main text.

Modifications of the manuscript:

We have added the following text to the main text (page 8 lines 142-148):

On the other hand, NTE (Neural-Surrogate-Guided Tree Exploration) is inherently a frequentist's approach and employs the number of visits to facilitate the exploration-exploitation trade-off. Unlike traditional Bayesian black-box optimization algorithms, which primarily use uncertainty as the basis for this trade-off, NTE treats the number of visits to a particular state as a measure of uncertainty. The more frequently a state is visited, the lower its associated uncertainty. This approach is common in Monte Carlo tree search-based methods. We have made some key modifications that deviate from traditional settings, enhancing our methodology's effectiveness.

'- That said, a key question arises:

What specifically makes DANTE superior to BO in the presented experiments? Is it the fact that the DNN surrogate model is more expressive compared to a GP? Or are the performance gains mainly attributed to the tree search algorithm to effectively navigate the candidate space. In my opinion, the latter would be significantly more impactful, and would require a key experiment involving a BO workflow that uses the exact same DNN model as DANTE as the surrogate. Approximate Bayesian inference methods (e.g. Laplace approximations or last-layer variational inference) should be readily available for neural networks. From this experiment, the contribution of the tree search method should be estimated. The response letter and the Supplementary Materials briefly mention a "CNN-BO" experiment, but due to insufficient details provided, it is difficult to assess precisely what has been done. This experiment, however, would be crucial for understanding the method's novelty and its primary contribution'

Response:

We thank the reviewer for the insightful questions and suggestions, which we address in the following. We fully comply and performed three more benchmark tests with 4 different combinations:

- 1) DNN + BO
- 2) DNN + DANTE
- 3) GP + DANTE
- 4) GP + BO

Figure R1. Benchmark performance of DANTE and BO with different surrogate models. Test functions are Ackley-20d; Rastrigin-20d; and Rosenbrock-20d. Selected surrogate models: Gaussian process (GP) and deep neural network (DNN). DANTE with DNN as surrogate model demonstrates a clear advantage over the other algorithms tested.

Figure R1 demonstrates that the DNN+DANTE combination achieves superior performance, exhibiting global convergence across all cases with a minimal number of data points. While GP+BO performs reasonably well across all scenarios, reaffirming that this classical combination remains a viable option alongside DANTE, both GP+DANTE and DNN+BO show relatively poor performance. Our results indicate that GP is rather good at approximating a low-dimensional landscape, while the DNN is more capable of approximating high-dimensional ones. On the other hand, BO performs well at the low dimensional case while DANTE excels at searching the optimum of this approximate high-dimensional landscape.

Again, the results suggest that GP+BO performs well in low-dimensional settings, whereas DNN+DANTE excels at navigating and locating optima within approximately high-dimensional landscapes. **These findings emphasize that selecting the most suitable pairing of surrogate and search models—based on the problem's dimensionality and nonlinearity—is crucial for achieving optimal overall performance.**

Regarding the search mechanism itself, a detailed ablation study shown in main text and in Figure 3b demonstrates that the backpropagation mechanisms and DUCB play critical roles in the SOTA performance of DANTE.

The evidence suggests that it is the combination of DNN and DANTE that makes our method particularly effective in the case of high-dimensional problems, and we further suggest that a more

powerful and expressive surrogate model at higher dimensions might help to solve even higher dimensional problems. However, we believe that this projection goes beyond the current scope of this paper.

Regarding further implementation details of CNN-BO we can add the following workflow information: We developed CNN-BO by modifying the source code of Bayesian Optimization (GitHub repository: <https://github.com/bayesian-optimization/BayesianOptimization>). Specifically, we replaced the original Gaussian Process Regressor with our CNN Regressor while retaining all other components of the codebase. The complete implementation of CNN-BO is publicly available at: https://github.com/Bop2000/DANTE/tree/main/CNN_BO.

Regarding the CNN Regressor architecture: We employ an ensemble of five convolutional neural networks (CNNs) with cross-validation to obtain predictive means and standard deviations. To enhance computational efficiency – as adopting the identical CNN architecture used in DANTE would incur prohibitive computational costs – we implemented a comparatively simpler 1D-CNN architecture. The 1D-CNN comprises 3 convolutional layers with filter sizes of 16, 8, and 4 respectively, each using a kernel size of 3. It also includes 1 max-pooling layer with a pooling size of 2 after the 1st convolutional layer, 1 dropout layer with a dropout rate of 0.2 after the max-pooling layer, followed by a flatten layer, 1 fully connected layer with 32 units, and an output layer. The loss function utilized is the mean square error (MSE). Moreover, the learning rate for the Adam Optimizer is set at 0.001, and the activation function utilized is the Exponential Linear Unit (ELU). The 1D-CNN model is trained for 100 epochs with a batch size of 32.

Notably, during the Ackley-20d benchmark test under identical hardware configurations, the CNN-DANTE approach successfully identified the global optimum in less than 10 minutes after acquiring 400-500 data points. In contrast, CNN-BO required over 24 hours to evaluate 1600 data points yet converged only to a suboptimal value of approximately 6. These results demonstrate CNN-DANTE's significant superiority over CNN-BO in both optimization efficiency and effectiveness.

Modifications of the manuscript:

1. We added the following text to the main text (page 15 lines 281-285):

Overall, the test results presented in Figure 3 (a and e) and Extended Figure 1 show that BO performs well in low-dimensional settings, whereas DANTE excels at navigating and locating optima within approximately high-dimensional landscapes. These findings emphasize that selecting the most suitable pairing of surrogate and search models—based on the problem's dimensionality and nonlinearity—is crucial for achieving optimal overall performance.

2. We added the following text to the main text, Extended Figure 1:

Extended Figure 1 (Main text). Benchmark performance of DANTE and BO with different surrogate models. Test functions are Ackley-20d; Rastrigin-20d; and Rosenbrock-20d. Selected surrogate models: Gaussian process (GP) and deep neural network (DNN). DANTE with DNN as surrogate model demonstrates a clear advantage over the other algorithms tested.

3. We added the following text to the Supplementary Information (page 6 lines 84-99):

We further tested a BO variant that employs CNNs as surrogate models. For the implementation details of CNN-BO: We developed CNN-BO by modifying the source code of Bayesian Optimization (GitHub repository: <https://github.com/bayesian-optimization/BayesianOptimization>). Specifically, we replaced the original Gaussian Process Regressor with our CNN Regressor while retaining all other components of the codebase. The complete implementation of CNN-BO is publicly available at: https://github.com/Bop2000/DANTE/tree/main/CNN_BO. The results of CNN-BO are shown in Supplementary Fig. 49-51, and Extended Figure 1. Regarding the CNNRegressor architecture: We employ an ensemble of five convolutional neural networks (CNNs) with cross-validation to obtain predictive means and standard deviations. To enhance computational efficiency – as adopting the identical CNN architecture used in DANTE would incur prohibitive computational costs – we implemented a comparatively simpler 1D-CNN architecture. The 1D-CNN comprises 3 convolutional layers with filter sizes of 16, 8, and 4 respectively, each using a kernel size of 3. It also includes 1 max-pooling layer with a pooling size of 2 after the 1st convolutional layer, 1 dropout layer with a dropout rate of 0.2 after the max-pooling layer, followed by a flatten layer, 1 fully connected layer with 32 units, and an output layer. The loss function utilized is the mean square error (MSE). Moreover, the learning rate for the Adam Optimizer is set at 0.001, and the activation function utilized is the Exponential Linear Unit (ELU). The 1D-CNN model is trained for 100 epochs with a batch size of 32.

Minor comments and remarks:

,

- Especially in the introduction, the use of the term "dataset" is somewhat confusing, especially given that the data continually changes in an iterative workflow. To improve clarity, clear distinctions should be made between initially available data and newly acquired data throughout the iterations.'

Response:

We thank the reviewer for the suggestions. We have revised the term "dataset" by the term "initial dataset" wherever appropriate to improve clarity.

Reviewer #2 (Remarks to the Author):

'The authors partially addressed my concerns. Especially for the high dimension problem, they presented two examples to show that high dimension exists. However, this is not general for real cases in materials science or alloy design.'

Response:

We contend that numerous problems in the scientific domain require high-dimensional optimization methods, as demonstrated by the newly added case studies in biology in our last round of response. For example, protein design problems can involve potential dimensions up to 1000. In this context, we assert that our work effectively demonstrates the practical value of the DANTE algorithm across eight diverse real-world tasks spanning the fields of physics, computer science, and biology.

Additionally, as we are engaged in other projects using DANTE, we would like to mention that we have a recent initiative involving a problem with 6,400 input variables. This project focuses on designing a specific type of Anderson Localization by optimizing the disorder matrix, providing an excellent illustration of high-dimensional optimization techniques applied to real-world cases.

'For example, the ice structure optimization, it is not a feasible way to use as many descriptors as possible as inputs, in particular to the materials or physics scientists. In general, the design of descriptors, such as order parameters based on expertise, is important in such studies and no body will want to collect or use too many descriptors, which complicates the problem concerned and makes the surrogate model or optimization a whole black-box. Using descriptors that are not quite important seems like "Grasp the shadow and lose the substance". If such idea works, then for materials or alloys design we can easily generate descriptors more than 1000, but I am not sure why we should do this.'

Response:

We thank the reviewer for the comment. We would like to humbly clarify that this paper is not primarily focused on the selection of descriptors for computational materials design tasks. Instead,

our main ambition and discussion centers on exploring the potential for advancing optimization techniques, with computational materials design serving as one of several application domains illustrating the work's broader objective.

On the other hand we sure fully agree with the reviewer that, for computational materials scientists and physicists, efforts to identify the most important descriptors often aim to gain physical insights into the properties of interest. In our example regarding ice structure optimization, the primary motivation for using high-dimensional inputs is to accurately reconstruct the ice structure through techniques such as reverse Monte Carlo (RMC). Specifically, the intricate structural complexity of ice leads to significant features in its radial distribution function (RDF) and angular distribution function (ADF) over long ranges. To capture these features with high fidelity, it is necessary to discretize the RDF and ADF more densely, as it is not possible to a priori determine the specific neighbor distances or angles at which these functions provide the most representative information.

Finally, the above argument also applies to other practical applications of the algorithm proposed in our work, particularly when domain knowledge is limited. In cases involving experimental data and the majority of data-driven materials design problems, noise is often substantial, and underlying theoretical models may be absent. Under such conditions, reliable feature analysis becomes challenging, and the common strategy is to utilize all available designable parameters to maximize the informational content and improve optimization outcomes.

'Also, for the implant design, is there some rules such as periodicity that can be used to simplify the optimization? The dimensions should not merely rely on the size of implant itself that cannot be simplified, just like the structure optimization of component in aerospace, which is even complex than the case here.'

Response:

Thank you very much for your valuable comment. We fully agree with the reviewer that implant optimization can benefit from simplification strategies, much like those used in aerospace structural design ¹; In current implant design practice, there are mainly three widely adopted approaches:

1. **Global topology optimization** of the entire structure
2. **Periodically arranged lattice designs**, typically using a single unit cell repeated throughout the domain;
3. **Graded or heterogeneous unit-based design**, where each local unit can be independently modified.

As the reviewer rightly pointed out, the first two strategies are effective in reducing the dimensionality of the design space by allowing optimization over a reduced set of parameters or

exploiting symmetry and periodicity. These methods are efficient and particularly suitable when only a single mechanical target (e.g., stiffness or mass) needs to be controlled.

However, in the context of **mechanically adaptive orthopedic implants**, especially those aiming to match both elastic modulus and strength to native bone, the performance requirements are inherently multi-objective. To address this, our study adopts the **third strategy**, where the scaffold is composed of a multi-array of TPMS-based gyroid units. This **unit-wise optimization approach** increases the dimensionality of the problem, as each unit becomes an independent variable in the design space. However, it allows **precise, localized control of multiple mechanical properties**, enabling the design of implants that meet complex clinical needs.

This strategy, while computationally more intensive, leads to significantly enhanced performance, as demonstrated in this study. Our previous work² has also validated this approach experimentally, showing that unit-wise optimized implants outperform uniform or topology-optimized counterparts in both mechanical behavior and clinical adaptability.

‘The authors also argue that more descriptors can improve the recovery capability. This may be correct, but there can be only a little decrease in recovery when many unimportant descriptors are removed. Just like the influence of descriptors dimension on the accuracy of surrogate model, more descriptors sometimes cannot increase but decrease the accuracy. In addition, I am wondering that for a certain problem, how the dimension change of descriptor affects the optimization efficiency? Because the dimension might change the distribution of the search space.’

Response:

We thank the reviewer for the insightful suggestions and we fully comply. As the reviewer correctly pointed out, changes in the descriptor dimension can indeed affect both the dimensionality and the nonlinearity of the problem, consequently transforming the optimization landscape.

To better understand the relationship between 1) optimization efficiency and problem dimension, and 2) optimization efficiency and the degrees of freedom (DoF) per dimension, we conducted two additional tasks using benchmark test functions. These analyses aimed to explore how these factors influence the optimization process in more detail.

Figure R2 demonstrates the influence of changes in the Degree of Freedom of the descriptor to the convergence behavior (Ackley-20d, Ackley-100d, Rastrigin-20d, Rastrigin-100d, Rosenbrock-20d, Rosenbrock-100d). As observed, as the degrees of freedom (DoF) increase, DANTE remains capable of achieving convergence, although it requires a larger amount of data. A heuristic relationship can be summarized as follows:

$$\text{Number of data points needed to reach the optimum} \approx c * \log(d * \text{DoF}),$$

where c is a constant and d represents the dimension. However, this formula does not explicitly account for the complexity of the nonlinear landscape itself and should therefore be used with caution.

Figure R2: Ablation study of DANTE on synthetic functions with different intervals (changes in the descriptor’s degree of freedom) of descriptor. **(a)** Exact functions, **(b)** surrogate model predictions. Data are presented as mean values \pm SD, $n = 5$.

Furthermore, we present two case studies utilizing benchmark test functions, namely Ackley and Rastrigin. Figure R2 illustrates the relationship between problem dimensionality and the amount of data required to achieve the global optimum.

Finally, while we aspire to approach this problem in a more general manner, our experience

suggests that it remains highly case-specific in practical situations. This specificity arises mainly because: **1) it is challenging to estimate the optimization landscape a priori, and 2) in some cases, the dimension has a practical meaning that cannot be altered.** For instance, drastically modifying a protein sequence (dimension change) is often undesirable, whereas such changes might be acceptable in certain materials science problems. We have incorporated this discussion into the main text to clarify these considerations.

Figure R3 demonstrates the influence of change in the number of dimensions to the optimization efficiency on the Ackley-20d and Rastrigin-20d functions. It can be observed that the data needed to achieve an optimum scales linearly with dimensionality.

Figure R3: Dimensions versus the number of data points needed to achieve the optimum. We evaluated DANTE on Ackley and Rastrigin functions with different dimensions. Data are presented as mean values \pm SD, $n = 5$.

'The authors also states that the descriptors from feature engineering is often suitable to a specific property. I cannot fully agree with this, as many cases have shown that if the descriptors can be properly defined based on, for example, domain knowledge or theory, they can generalize to other properties well. This is underpinned by the robust generalizability of theory or knowledge, for example, the Hall-Petch equation.'

Response:

We agree with the reviewer that there are some well-understood problems with known foundations, however we should also agree that there are many other problems with unknown theory. This is particularly true in the experimental science. In cases involving experimental data and the majority of data-driven materials design problems, noise is often substantial, and underlying theoretical models may be absent (this often holds for high-entropy alloy design). Under such conditions, reliable feature analysis becomes challenging, and the common strategy is to utilize all

available designable parameters to maximize the informational content and improve optimization outcomes.

Modifications of the manuscript:

1. We added the following text to the Supplementary Information (page 8 lines 146 to 159):

Optimization efficiency versus dimension: To better understand the relationship between 1) optimization efficiency and problem dimension, and 2) optimization efficiency and the degrees of freedom (DoF) per dimension, we conducted two additional tasks using benchmark test functions. These analyses aimed to explore how these factors influence the optimization process in more detail.

Supplementary Figure 7 demonstrates the influence of changes in the Degree of Freedom of the descriptor to the convergence behavior (Ackley-20d, Ackley-100d, Rastrigin-20d, Rastrigin-100d, Rosenbrock-20d, Rosenbrock-100d). As observed, as the degrees of freedom (DoF) increase, DANTE remains capable of achieving convergence, although it requires a larger amount of data. A heuristic relationship can be summarized as follows:

$$\text{Number of data points needed to reach the optimum} \approx c * \log(d * \text{DoF}),$$

where c is a constant and d represents the dimension. However, this formula does not explicitly account for the complexity of the nonlinear landscape itself and should therefore be used with caution.

2. We add the following figures to the Supplementary Information (Supplementary Figure 53):

Supplementary Fig.53 demonstrates the influence of change of dimension to the optimization efficiency on Ackley and Rastrigin functions. It can be observed that the data needed to achieve optimum scales linearly with the dimensionality.

Supplementary Figure 53: Dimensions versus the number of data points needed to achieve the optimum. We evaluated DANTE on Ackley and Rastrigin functions with different dimensions. Data are presented as mean values \pm SD, $n = 5$.

References:

1. Aage, N., Andreassen, E., Lazarov, B. S. & Sigmund, O. Giga-voxel computational morphogenesis for structural design. *Nature* 550, 84–86 (2017).
2. Peng, B. *et al.* Machine learning-enabled constrained multi-objective design of architected materials. *Nat Commun* 14, 6630 (2023).

Response to reviewer's comments

Reviewer #1 (Remarks to the Author):

'However, the authors appear to have ignored my earlier concern regarding the use of the term "active learning", which was also expressed by other reviewers in previous rounds of review. In the field, "active learning" typically refers to iterative workflows aimed at sample-efficient data acquisition to improve model predictivity. In contrast, the approach in this work focuses on sample-efficient data acquisition for optimizing an unknown black-box function, which resembles the classical use case of BO. While these differences are subtle, they have a number of practical implications. Therefore, in my opinion, it is important to make this distinction, in order to avoid potential misconceptions about the scope and objectives of the paper.'

Response:

Thanks for the reminder, to address the concern, We re-define our algorithm as "Active Optimization" (AO), which closely aligns with Bayesian Optimization (BO) in terms of its objectives and overarching framework. However, while BO primarily relies on kernel-based surrogate models and uncertainty-driven acquisition functions to select optimal candidates, AO extends this paradigm by generalizing the application of surrogate modeling and search strategies.

This broader approach allows AO to adapt across a diverse array of method types, thereby increasing its versatility and scope beyond conventional BO methodologies. Additionally, AO shares similarities with active learning (AL) frameworks but diverges in its primary goal; rather than solely improving model predictive accuracy, AO seeks to identify optimal solutions starting from a relatively small initial dataset—ranging from a few dozen to hundreds of data points. Like BO and AL, AO comprises two principal components, in addition to the validation source and database, which underpin its operational structure.

Modifications:

We change the article title to:

Deep active optimization for complex systems

We add the following text to the main text (line 40-48):

As illustrated in Figure 1, We designate our algorithm as 'Active Optimization' (AO), which aligns closely with Bayesian Optimization (BO) in terms of its objectives and overall framework. However, BO primarily utilizes kernel method and uncertainty-based acquisition function to identify 'optimal' candidates, whereas AO generalizes the application of surrogate models and search methodologies, allowing for adaptation across a wider variety

of method types, thereby enhancing its versatility and scope beyond traditional BO approaches. Furthermore, AO is akin to the AL framework but differs in terms of its goal - instead of improving the model predictivity, AO aims at finding the optimal solutions with a relatively small initial dataset (from a few dozen to hundreds).